# Towards Safe Machine Unlearning: a Paradigm that Mitigates Performance Degradation

## Abstract

We study the machine unlearning problem which aims to remove specific training data from a pre-trained machine learning model to allow users to exercise their 'right to be forgotten' to protect user privacy. Conventional machine unlearning methods would degrade the model performance after the unlearning procedure. To mitigate the issue, they typically rely on the access to the remaining training data to fine-tune the unlearned model to mitigate the influence of unlearning. However, accessing the remaining training data may not always be practical for different reasons (e.g., data expiration policies, storage limitations, or additional privacy constraints). Machine unlearning without access to the remaining training data poses significant challenges to retaining model performance. In this paper, we study how to unlearn specific training data from a pre-trained model without accessing the remaining training data and protect model performance without dramatically changing the model's parameters. We propose a practical method called Targeted Label Noise Injection. Intuitively, our method assigns incorrect yet controllable labels to the examples that need to be forgotten and fine-tunes the pre-trained model to learn these new labels. This strategy effectively moves the to-be-forgotten examples across the decision boundary with a small impact on the model's overall performance. We theoretically prove the effectiveness of the proposed method and empirically show that it achieves state-of-the-art unlearning performance across various datasets.

**Relevance**: Machine unlearning is critical to user modeling because users have the 'right to be forgotten'. This paper proposes a novel paradigm for machine unlearning to mitigate model performance degradation during unlearning, which is highly relevant to the track of 'User modeling, personalization and recommendation'. Moreover, the proposed method provides new state-of-the-art for Web applications where both high privacy and utility are required.

## CCS Concepts

• **Networks** → **Network privacy and anonymity**; • **Security and privacy** → *Usability in security and privacy*.

## Keywords

Privacy protection, machine unlearning, trustworthy machine learning

**ACM Reference Format:**
Anonymous Author(s). 2018. Towards Safe Machine Unlearning: a Paradigm that Mitigates Performance Degradation. In *Proceedings of the ACM Web Conference 2025 (WWW'25), April 28th–May 2nd, 2025, Sydney, NSW, Australia.* ACM, New York, NY, USA, 23 pages. https://doi.org/XXXXXXX.XXXXXXX

## 1 Introduction

In the era of big data, machine learning models have become integral to a wide range of web applications, from personalized recommendations to critical decision-making systems. These models are often trained on vast amounts of user data, much of which may be sensitive or personal. As concerns about data privacy escalate, stringent regulations such as the General Data Protection Regulation (GDPR) [27] have been enacted, granting individuals the right to have their personal data erased/forgotten. There is a growing need for machine learning models to not only learn from data but also to unlearn specific data points upon request.

Machine unlearning [2] aiming to remove specific data from trained machine learning models has thus emerged as a crucial capability. This ensures compliance with data privacy laws and maintains user trust in systems that handle personal information. Unfortunately, the unlearning procedure would degrade model performance. To address this issue, conventional methods typically require access to both the data to be forgotten and the remaining training data, using the remaining data to tune the unlearned model to mitigate performance degradation [1, 5, 35]. However, accessing the remaining data can be impractical due to many reasons, e.g., data expiration policies that mandate deletion after a certain period, storage limitations that prevent retaining large datasets, or additional privacy constraints that prohibit the reuse of data without explicit consent.

Unlearning without access to the remaining data poses significant challenges. Existing machine unlearning methods in this setting (e.g., [4, 8, 23]) are limited and often lead to substantial changes in the model's parameters, which can degrade performance on the remaining data that should be retained. Intuitively, fine-tuning solely on the examples to be forgotten may require extensive parameter adjustments, resulting in overfitting the to-be-forgotten examples and underperformance on the remaining examples. Motivated by these challenges, in this paper, we study how to safely unlearn some examples from a pre-trained model without accessing the remaining training examples and without dramatically changing the model's parameters.

To address this problem, we propose a novel method called Targeted Label Noise Injection for machine unlearning. The key philosophy of our method is to minimize parameter changes by selecting incorrect yet easy-to-learn labels for the to-be-forgotten examples. Specifically, we assign each instance to be forgotten an incorrect label that the model is already somewhat inclined toward. This ensures that the pseudo-label aligns relatively well with the model's existing knowledge and is easy for the model to learn. By doing

so, we reduce the need for large parameter updates, helping to maintain the model's performance on the remaining examples.

However, it is also important to note that during fine-tuning, the model can overfit the pseudo-labels with unnecessarily high confidence. This leads to unwanted parameter adjustments. To prevent overfitting the pseudo-labels, we reduce the contribution of each to-be-forgotten example to the loss once the model starts predicting its pseudo-label correctly. This is achieved by introducing an exponent $\lambda$ to the training loss. By setting a high value for $\lambda$ during the fine-tuning process, the loss of instances with pseudo-labels, which the model has already learned, is dramatically reduced. Consequently, in the later fine-tuning epochs, these instances have little influence on the model's parameter updates, effectively avoiding unnecessary changes to the model's parameters.

Furthermore, we theoretically analyze the parameter changes induced by our method and find that it concentrates updates on a specific subset of parameters associated with to-be-forgotten examples. This targeted adjustment effectively modifies only the necessary parameters while minimizing the impact on unrelated parts of the model. Empirically, we have validated that our method achieves the smallest parameter changes across different datasets when compared with various baselines. Our method consistently achieves state-of-the-art performance for machine unlearning without accessing the remaining data, demonstrating its effectiveness and practicality across diverse scenarios.

## 2 Related Work

We study the machine unlearning aiming to enable a pre-trained model to discard information acquired from a subset of data. The primary objective has been to undo the influence of undesired data on the model while maintaining the model's predictive power on the remaining sample. Historically, simpler machine learning models, such as linear/logistic regression, k-means clustering, and random forests, have been the subjects of various unlearning techniques (e.g., [10, 24, 25]). However, these methodologies are inherently designed for these less complex models and don't seamlessly translate to intricate architectures like deep neural networks.

To achieve unlearning for deep neural networks, many methods have also been proposed. Some of them focus on making the model forget all examples corresponding to a specific class, like all images of a particular category in a dataset [5, 35, 36]. Some of them emphasize erasing information from some examples potentially spanning multiple classes from the pre-trained model [3, 26, 29]. By considering that data deletion may requests are practically received on a per-instance basis, some methods which allow to forget any subset of training examples have also been proposed [9, 13, 37]. A notable limitation shared by most method is their reliance on access to both the data intended for unlearning and insights on the remaining training data during the unlearning phase (e.g., [17, 21, 31]). This dependency becomes problematic in real-world scenarios where accessing the remaining sample might be hindered by factors like data expiration policies or other regularization.

To circumvent this challenge, the use of gradient ascent in the unlearning process has been naturally proposed. The basic idea is to employ gradient ascent to reverse the loss value of an example marked for unlearning from a pre-trained model. However, while the gradient ascent-based unlearning method has demonstrated its effectiveness across diverse settings and datasets [16, 28, 33], concerns have been raised about its suitability in scenarios demanding high privacy and utility. Graves et al. [12] argued that merely amplifying loss using gradient ascent doesn't inherently ensure heightened privacy. Suriyakumar and Wilson [30] further highlighted the potential for data leakage even when the loss is accentuated. This brings to light the privacy implications of deploying existing techniques. Some methods have also been proposed which use gradient descent (e.g., [16, 22, 32]). However, fine-tuning solely on the examples to be forgotten may require extensive parameter adjustments. This leads to overfitting to the to-be-forgotten examples and underperformance on remaining examples. Motivated by these challenges, we study how to safely unlearn examples from a pre-trained model without accessing the remaining training examples and without dramatically changing the model's parameters.

## 3 $\delta$-Targeted$^\lambda$ Label Noise Injection for Machine Unlearning

**Problem Setup.** Consider a dataset $D = \{(x_i, y_i)\}_{i=1}^n$, where $x_i \in \mathcal{X}$ is an instance from the feature space, $y_i \in \{1, 2, \ldots, C\}$ is its corresponding label from $C$ classes, $n$ is the size of the dataset. Each example $(x_i, y_i)$ is assumed to be independently drawn from an underlying distribution $P(X, Y)$.

Let $\sigma \circ f_{\hat{\theta}} : \mathcal{X} \to \mathbb{R}^C$ be a pre-trained model trained on $D$ by minimizing the average loss, where $f_{\hat{\theta}}$ represents the deep network and $\hat{\theta}$ denotes the trained parameters, and $\sigma$ is the softmax function. Given an instance $x$, the model outputs $\sigma \circ f_{\hat{\theta}}(x)$ estimates the class-posterior distribution.

Suppose we have a subset $D_f \subseteq D$ of examples that we aim to forget from the model. The remaining examples are denoted as $D_r = D \setminus D_f$. Our objective is to obtain a model that effectively forgets $D_f$ while maintaining high accuracy on $D_r$ and generalization, using only $D_f$ and the pre-trained model $\sigma \circ f_{\hat{\theta}}$.

### 3.1 Targeted Label Noise Injection with Probability $\delta$

Our method for machine unlearning is based on injecting label noise into the to-be-forgotten examples. Intuitively, to mitigate performance degradation, we select incorrect yet easy-to-learn labels for the to-be-forgotten examples. This approach ensures that the pseudo-label aligns relatively well with the model's existing knowledge and is easy to learn. This effectively reduces the need for large parameter changes and helps to maintain the model's performance.

Specifically, for each to-be-forgotten example $(x, y) \in D_f$, we compute the model's predicted probabilities $\sigma \circ f_{\hat{\theta}}(x)$. Let $\hat{y} = \arg\max_{i \in \{1,2,\ldots,C\}} \sigma \circ f_{\hat{\theta}}(x)$ be the predicted label. The second most confident label is defined by $\hat{y}' = \arg\max_{i \in \{1,2,\ldots,C\} \setminus \hat{y}} \sigma \circ f_{\hat{\theta}}(x)$. It is easy to find that the second most confident label of a to-be-forgotten example is an easy-to-learn label for it. In this paper, we use the second most confident labels as the easy-to-learn labels.

By considering that different scenarios may require different degrees of forgetting, we might want to control the trade-off between

the level of unlearning and the preservation of the models's performance. Additionally, if all to-be-forgotten examples are always assigned incorrect labels, an adversary could infer that the true labels are different from the predicted ones, thus gaining information about the original data. Therefore, we introduce a parameter $\delta \in [0, 1]$ to control the strength of the label perturbation. Specifically, if the model's predicted label $\hat{y}$ does not match the original label $y$, this indicates a misprediction. In such a case, with a probability of $1 - \delta$, the pseudo label $y'$ is set to the predicted label $\hat{y}$; and with a probability of $\delta$, the original label $y$ is retained as the pseudo label $y'$. Conversely, if the predicted label $\hat{y}$ matches the original label $y$, indicating a correct prediction, then with a probability of $1 - \delta$, the pseudo label $y'$ is set to the second most confident label $\hat{y}'$; and with a probability of $\delta$, the original label $y$ is retained as the pseudo label $y'$. Formally, this assignment of the pseudo label $y'$ can be expressed as:

$$y' := \begin{cases} y, & \text{if } \alpha < \delta; \\ \hat{y}', & \text{if } \hat{y} = y \wedge \alpha \geq \delta; \\ \hat{y}, & \text{if } \hat{y} \neq y \wedge \alpha \geq \delta, \end{cases}$$

where $\alpha \sim \text{Uniform}(0, 1)$ is a random variable. By choosing a smaller $\delta$, we enforce that more pseudo-labels differ from the original labels. To let the pre-trained model randomly guess a to-be-forgotten example, $\delta$ should be set to $1/C$, where $C$ represents the number of unique labels.

## 3.2 Avoid Overfitting to Label Noise with a Loss Exponent $\lambda$

After injecting label noise into the to-be-forgotten examples. A set of new to-be-forgotten examples $D'_f = \{x_i, y'_i\}_{i=1}^m$ are obtained which contains instances and their pseudo labels[1], where $m$ is the size of example to be forgotten. The unlearning process aims to minimize the loss induced by the pseudo labels while preserving the accuracy on the remaining sample $D_r$ by only the sample $D_f$. Note that since the pseudo labels are mostly different from the original labels, small losses with respect to the pseudo labels imply large losses with respect to the original labels. Examples with large loss values mean that they have not been fitted by the model and are thus unlearned.

However, we notice that minimizing the loss natively might lead the model to fit the pseudo labels with unnecessarily high confidence, which is not our primary objective. Our goal is to make the model mispredict the original labels of these instances using pseudo labels, but without letting the model overfit these pseudo labels with high confidence. This distinction is crucial. If the model already predicts a pseudo label for a to-be-forgotten example correctly, there's no need to adjust its parameters further to increase the confidence of this prediction. Instead, during the unlearning procedure, the model updating should focus on the pseudo labels that it cannot predict correctly. By doing so, we can avoid unnecessary parameter adjustments, which could lead to overfitting the to-be-forgotten examples and adversely affect the model's performance on remaining examples.

---

[1]Note that in the rest of the paper when there is no confusion, we do not distinguish to-be-forgotten examples with original labels and pseudo label.

**Algorithm 1** $\delta$-Targeted$^\gamma$ Label Noise Injection For Machine Unlearning

1: **procedure** UNLEARN($D_f, \hat{\theta}, \eta, \delta$)
2:     $D'_f \leftarrow \emptyset$ ▷ Initialize an empty set
3:     **for** each $(x, y) \in D_f$ **do** ▷ Iterate over all examples in the to-be-forgotten examples
4:         $\hat{y} \leftarrow \arg\max_i \sigma \circ f_{\hat{\theta}}(x)$ ▷ The model's predicted label
5:         $\hat{y}' \leftarrow \arg\max_{i \neq \hat{y}} \sigma \circ f_{\hat{\theta}}(x)$ ▷ The model's second confident label
6:         $\alpha \sim \mathcal{U}(0, 1)$ ▷ Draw a scalar from the uniform distribution.
7:         **if** $\hat{y} \neq y \wedge \alpha < \delta$ **then**
8:             $y' \leftarrow y$ ▷ Retain the original label
9:         **else if** $\hat{y} \neq y \wedge \alpha \geq \delta$ **then**
10:             $y' \leftarrow \hat{y}$ ▷ Let the pseudo label be the model's predicted label.
11:         **else if** $\hat{y} = y \wedge \alpha < \delta$ **then**
12:             $y' \leftarrow y$ ▷ Retain the original label
13:         **else**
14:             $y' \leftarrow \hat{y}'$ ▷ Let the pseudo label be the model's second confident label.
15:         **end if**
16:         $D'_f \leftarrow D'_f \bigcup \{(x, y')\}$ ▷ Add new example $(x, y')$ to $D'_f$
17:     **end for**
18:     $\hat{\theta}'' \leftarrow \hat{\theta}$ ▷ Initialize the parameter needed to be fine-tuned
19:     **while** $L'(\hat{\theta}'', D'_f) \geq \epsilon$ **do**
20:         Compute $\nabla L'(\hat{\theta}'', D'_f)$ using Eq. (1) ▷ Compute gradient of modified loss
21:         $\hat{\theta}'' \leftarrow \hat{\theta}'' - \eta \nabla L'(\hat{\theta}'', D'_f)$ ▷ Update the model's parameters
22:     **end while**
23:     **return** $\hat{\theta}''$ ▷ Return the updated model's parameters
24: **end procedure**

A natural approach is to dynamically remove examples from the training set once their pseudo labels can be accurately predicted by a model during the unlearning process. This ensures that these examples are not further learned. The model should then concentrate on learning the pseudo labels of the remaining to-be-forgotten examples that it cannot predict accurately. However, we found that this should not work well in practice. The primary issue is that learning the pseudo labels for the remaining examples alters the model's parameters. After these changes, the model may not remember the pseudo labels of previously removed examples from the to-be-forgotten examples. Therefore, instead of removing an example from the training set once its pseudo label can be accurately predicted, a more effective method is to reduce the example's contribution to the optimizing objective if the example already exhibits a small loss.

Motivated by this, we propose using an exponent $\gamma$ to control the loss contribution of to-be-forgotten examples during the learning process. If the model already aligns well with the pseudo label of a to-be-forgotten example, instead of removing this example from the training set, we reduce its contribution to the loss with an exponent $\gamma$, i.e., $\ell^\gamma$. Note that if $\gamma > 1$, when $\ell$ is close to zero, $\ell^\gamma$ will become

smaller than $\ell$; while when $\ell$ is larger than one, $\ell^\gamma$ will become larger than $\ell$. If the model predicts the pseudo label of a to-be-forgotten example, the corresponding loss value should be close to zero. If the model struggles to predict the pseudo label of a to-be-forgotten example, that example will have a larger loss compared with other examples whose labels are correctly predicted, encouraging the model to focus more on learning such an unfitted example. This method automatically adjusts the influence of each to-be-forgotten example based on its prediction accuracy. By decreasing the loss contribution from well-predicted examples and increasing it for poorly predicted ones during the unlearning process, this method ensures more efficient and targeted unlearning.

Formally, we exponentiate cross-entropy loss function $\ell(\sigma \circ f_\theta(\boldsymbol{x}), \boldsymbol{y}')$ by a exponent $\lambda$, where $\boldsymbol{y}'$ is the one-hot encoding form of pseudo label $y'$. This implicitly introduces a dynamic weight associated with each example. The objective function is defined as:

$$\arg\min_\theta L'(\theta, \boldsymbol{D}'_f) = \arg\min_\theta \frac{1}{m} \sum_{(\boldsymbol{x}, \boldsymbol{y}') \in D'_f} \frac{\ell(\sigma(f_\theta)(\boldsymbol{x}), \boldsymbol{y}')^\lambda}{m}. \quad (1)$$

By raising the loss to the power of $\lambda$, we dynamically adjust the contribution of each instance to the overall objective. This method ensures that the model concentrates on learning the pseudo-labels it hasn't yet mastered, without over-adjusting parameters for those it already predicts correctly. By preventing overfitting to the pseudo-labels, we avoid unnecessary parameter changes that could degrade performance on the remaining data. The gradient update during fine-tuning is then

$$\theta \leftarrow \theta - \eta \nabla L'(\theta, \boldsymbol{D}'_f), \quad (2)$$

where $\eta$ is the learning rate. By combining the above with the targeted label noise injection, our unlearning strategy effectively reduces the risk of over-adjusting the model parameters during the fine-tuning process. We name our method $\delta$-Targeted$^\gamma$ Sample Unlearning, the pseudo-code is illustrated in Algorithm 1.

## 4 Theoretical Analysis

In this section, we provide a theoretical analysis demonstrating how our method concentrates parameter updates on the specific subset of parameters associated with the to-be-forgotten examples. This targeted adjustment effectively modifies the necessary parameters while reducing the impact on unrelated parts of the model. We compare our method with sample unlearning by applying gradient ascent on original labels. Note that gradient ascent is commonly employed by existing methods (e.g., [11, 22, 32]). In experiments, we show that benefiting from the concentrated parameter updates, our method leads to smaller changes in parameters compared with all baselines on different neural network structures. As a consequences, the smaller change in parameter further lead to the small performance degradtion

We formally introduce Concentrated Parameter Updates in the following definition.

DEFINITION 4.1 (CONCENTRATED PARAMETER UPDATES). *Let $\theta$ denote the original parameter set of a pre-trained model, and let $\theta_1$ and $\theta_2$ be two distinct sets of updated parameters. Define the changes in parameters for $\theta_1$ and $\theta_2$ relative to $\theta$ as $\Delta\theta_1 = \theta_1 - \theta$ and $\Delta\theta_2 = \theta_2 - \theta$, respectively. We say that $\theta_2$ represents a more concentrated parameter update than $\theta_1$ if*

*(1) $\|\Delta\theta_1\|_1 = \|\Delta\theta_2\|_1$, and*
*(2) $\Delta\theta_2$ exhibits larger changes for certain parameters and smaller influence for others, compared to a more uniformly distributed change in $\Delta\theta_1$.*

Note that the above definition restricts the total magnitude of change $\|\Delta\theta_1\|_1 = \|\Delta\theta_2\|_1$ across two sets of parameters. It ensures that comparisons are not biased by the scale of the changes, thereby facilitating a fair comparison of parameter updates based on their distribution of changes.

Given the non-convex optimization landscape of neural networks and their inherent non-linearity, a fully transparent statistical analysis comparing the effects of gradient ascent and descent to all parameters is ambitious. Empirically, we have found that changes in the last fully connected layer serve as an effective surrogate for monitoring changes in the model's parameters. Specifically, let denote $\phi$ as the parameter of the last fully connected layer, in Section 5.1, we demonstrate that variations in $\phi$ exhibit a strong dependence with changes spanning all parameters[2]. Hence, we narrow our analysis to examine the impact of the unlearning process on the last fully connected layer's parameters.

In the following theorem, we show that after unlearning, our Targeted Sample Unlearning yields a more concentrated parameter update compared to the application of gradient ascent.

THEOREM 4.1 (PARAMETER UPDATES FOR $\delta$-TARGETED$^\lambda$). *Consider a pre-trained neural network model with parameters $\hat\theta = \hat\psi, \hat\phi$, where $\hat\psi$ represents the parameter of all layers except the last fully connected layer, $\hat\phi$ represents the parameter of the last fully connected layer, $C$ is the number of classes, and $M$ is the dimension of the feature representation. Let $\boldsymbol{o} = g_{\hat\psi}(\boldsymbol{x}) \in \mathbb{R}^M$ represent the activation before the last linear layer for an input $\boldsymbol{x}$, with all components $\boldsymbol{o}_i > 0$ (e.g., after ReLU activation). The logits are given by $f_{\hat\phi}(\boldsymbol{x}) = h_{\hat\phi}(\boldsymbol{o}) = \hat\phi\boldsymbol{o}$. We compare two updated parameter sets $\phi''$ and $\phi'$ obtained after unlearning a to-be-forgotten example $(\boldsymbol{x}, y)$:*
*1. Our $\delta$-Targeted$^\lambda$ Method Update (TD) with Pseudo-Label $y'$:*

$$\phi'' = \hat\phi - \eta\nabla_{\hat\phi}\ell(\sigma(h_{\hat\phi}(\boldsymbol{o})), \boldsymbol{y}') = \hat\phi - \eta(\sigma(h_{\hat\phi}(\boldsymbol{o})) - \boldsymbol{y}')\boldsymbol{o}^\top;$$

*2. Gradient Ascent Update (GA) with Original Label $y$:*

$$\phi' = \hat\phi + \eta\nabla_{\hat\phi}\ell(\sigma(h_{\hat\phi}(\boldsymbol{o})), \boldsymbol{y}) = \hat\phi + \eta'(\sigma(h_{\hat\phi}(\boldsymbol{o})) - \boldsymbol{y})\boldsymbol{o}^\top,$$

*where $\sigma$ is the softmax function, $\ell$ is the cross-entropy loss, $\boldsymbol{y}$ is the one-hot encoding of the original label, $\boldsymbol{y}'$ is the one-hot encoding of the pseudo-label assigned by our method, and $\eta$ and $\eta'$ are the learning rates. According to Definition 4.1, the parameter updates in $\phi''$ are more concentrated compared to those in $\phi'$. Specifically, the updates in $\phi''$ are concentrated on the weights associated with the pseudo-label $y'$ and the original label $y$, while the updates in $\phi'$ are distributed across all classes.*

**Proof Intuition.** In gradient ascent, the update direction is the gradient of the loss with respect to the original label $\boldsymbol{y}$. This gradient changes all parameters associated with not only the true class $y$ but also all other classes, leading to more uniformly distributed parameter changes. In our method, we perform gradient descent

---

[2]An intuitive explanation for this phenomenon is that changes to the last fully connected layer's parameters get amplified and propagated to former layers due to the chain rule employed during backpropagation.

on the loss with respect to the pseudo-label $y'$, which is the second most confident prediction. The gradient primarily affects the parameters associated with the pseudo-label $y'$ and the activations corresponding to $o$. Since $y'$ is chosen to be the class the model is already somewhat confident about, the parameter changes are concentrated on the weights connecting $o$ to $y'$, leading to more focused parameter updates. Therefore, under the condition that the total magnitude of changes is equal, our method results in parameter changes that are more concentrated on a subset of parameters, which satisfies the concentrated parameter update in Definition 4.1. Proof details can be found in the appendix.

Theorem 4.1 provides a theoretical justification for the $\delta$-Targeted$^\lambda$ method. It is important to note that for our theoretical analysis, we set both $\lambda = 1$ and $\delta = 1$. The parameter $\delta$ controls the privacy level. Setting $\delta = 1$ ensures that the model mispredicts labels for all to-be-forgotten examples after fine-tuning. We chose $\delta = 1$ specifically to analyze how parameter changes induced by the $\delta$-Targeted$^\lambda$ method compare to the method induced by a gradient ascent when a model is intensively fine-tuned to force misprediction of an example. Additionally, we employ a first-order cross-entropy loss. By increasing the order of the cross-entropy loss (i.e., setting $\lambda > 1$), the change to the model should be more focused. This is because higher values of $\lambda$ emphasize larger losses, causing the network to concentrate more on examples with large training losses. Empirically, results for $\lambda = 1$ and $\lambda = 3$ are presented in Section 5.1. The results are consistent where $\lambda = 3$ performs better in most cases when the size of to-be-forgotten examples is large.

## 5 Experiments

**Datasets and Baselines** The evaluation of our unlearning methods was conducted using four different benchmark image classification datasets: CIFAR-10, CIFAR-100 [18], Fashion-MNIST [34], and Tiny-ImageNet [20]. CIFAR-10 contains 10 classes with 50,000 training images and 10,000 test images. CIFAR-100 includes 100 classes with 50,000 training images and 10,000 test images. Fashion-MNIST has 10 classes with 60,000 training images and 10,000 test images. Tiny-ImageNet contains 200 classes with 100,000 training images and 10,000 test images.

We utilized ResNet-18, ResNet-50 [14] and vision transformer (ViT) architectures [7] for our experiments. We established a comparison of our proposed methods against different baselines. These include "*BEFORE*", which represents the state of the model before unlearning; "NegGrad", which involves fine-tuning models using gradient ascent [19]; "RandL", which misleads the model through random label perturbation [15]; "Bad Teacher", which uses competent and incompetent teachers in a student-teacher framework to induce forgetfulness [6]; "Amnesiac", which undoes the parameter updates from only the batches containing sensitive data [12]. Our proposed methods are $\delta$-Targeted$^1$, which sets the $\lambda = 1$ and $\delta$-Targeted$^3$, which sets $\lambda = 3$. Note that for "Bad Teacher" and "Amnesiac", some remaining examples need to be accessed. In our setting, we provide them with 300 remaining examples. For all other methods, no remaining examples are provided.

**Experiment Settings** For each dataset, a random subset of images from the training dataset was selected to form the unlearning sample, denoted as $D_f$. The size of this sample varied, with cardinalities

set at 4, 16, 64, 128, 256, 512, 1024, respectively. For our method, the unlearning process was carried out using an Adam optimizer. The parameters of the Adam optimizer were kept constant across all experiments, with a learning rate of $1e - 4$, a weight decay of $1e - 5$. To ensure statistical significance, all experiments were conducted five times, each with different random initializations for each setting. The standard deviation of the results from each five trials has also been reported.

Note that to make the pre-trained model randomly guess a to-be-forgotten example, $\delta$ should be set to $1/C$, where $C$ is the number of classes. In our experiments, we test our method under the most challenging conditions, we set $\delta = 0$ in our experiments. This requires the model to misclassify all to-be-forgotten examples by learning from the label noise while preserving performance on the remaining examples.

## 5.1 Magnitude of Parameter Changes for Different Methods

In Section 4, we showed that our method results in more concentrated parameter updates for the last-layer parameters than other methods. We hypothesize that concentrated updates in the last-layer parameters should lead to smaller changes in all parameters. We validate this on different neural network architectures, datasets, and baselines.

In Fig. 1, we measure and compare the parameter changes introduced by different unlearning methods, quantified using the $L_1$-norm. The results confirm that our proposed method consistently induces smaller parameter changes after unlearning. This empirical evidence supports our method's effectiveness in not only forgetting specific data but also in inducing only small changes to the model's parameters. Moreover, the consistent pattern of small parameter changes observed across various datasets and neural network architectures underscores the robustness of our method across different datasets and models.

## 5.2 Relations between Accuracies and Parameter Changes

In Fig. 2, we visualize the relationship between the change in all parameters of the model and test accuracy using scatter plots. The change in the models' parameters is quantified using the $L_1$-norm. The dependence between changes in last-layer parameters and model accuracy on remaining examples using our method is illustrated in App.B.9.

For each subplot, $\delta$-Targeted$^1$ and $\delta$-Targeted$^3$ are collectively displayed. A linear regression line obtained by regressing models' test accuracies against changes in parameters is also shown. The results indicate that there is a consistent negative dependence between the change in the models' parameters $\Delta\theta$ and test accuracy across various datasets. These findings suggest that small changes in parameters during the unlearning process contribute to maintaining test accuracy. This confirms that our method, by inducing small changes to the model's parameters, successfully maintains the performance of the model after unlearning.

Additionally, smaller models seem to be more sensitive to parameter changes, which impacts their generalization ability. By examining Fig. 2 and Fig. 7 in the appendix, it is evident that across

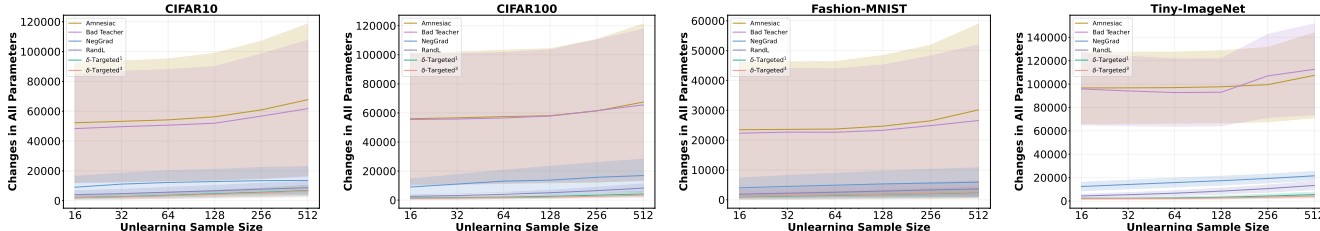

**Figure 1: Parameter changes for different numbers of to-be-forgotten examples across different datasets. The figure includes three neural network architectures: ResNet-18, ResNet-50, and ViT. The changes are quantified by the $L_1$-norm. Our method results in the smallest changes to the models' parameters across the datasets. Results for individual neural network architectures are provided in App.B.6.**

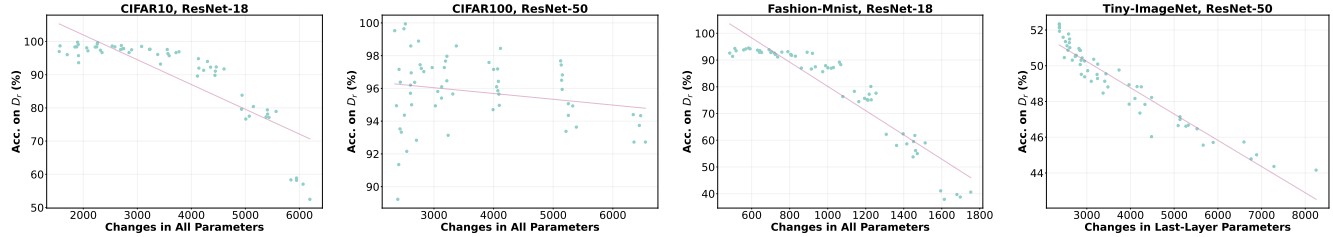

**Figure 2: Dependence between changes in all parameters and model accuracy on remaining examples by using our method. The result shows a negative correlation. The results show a positive correlation. Results for other neural network architectures are provided in App.B.8.**

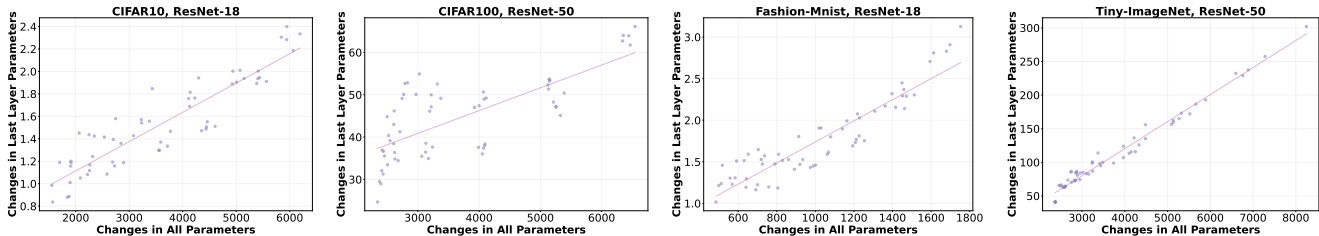

**Figure 3: Dependence between the changes in all parameters and the changes in the last layer parameters by using our method. The magnitude of the parameter changes is quantified by the $L_1$-norm. The results show a positive correlation. Results for other neural network architectures are provided in App.B.7.**

most datasets, the same amount of parameter change causes a larger drop in accuracy for smaller neural network models. Another interesting observation from the figure is that the rate at which accuracy decreases in response to parameter changes varies by dataset. The accuracy drop is fastest on Fashion-MNIST, followed by CIFAR-10, CIFAR-100, and Tiny-ImageNet. This trend may suggest that models trained on more complex tasks are not as vulnerable to losses in accuracy due to parameter changes.

### 5.3 Relations between Changes in Last-Layer Parameters and Changes in All Parameters

In Section 4, we showed that our method results in concentrated parameter updates in the last-layer parameters. We focused on parameter changes in the last layer instead of changes in all parameters of a model. We hypothesize that updates in the last-layer

parameters should lead to smaller changes in all parameters. We validate this on different neural network architectures and datasets in this subsection.

Fig. 3 demonstrates the relationship between changes in the last-layer parameters and changes in a model's overall parameters. Each subfigure represents a dataset and neural network architecture combination. The data reveal a consistent positive correlation between changes in the last-layer parameters and changes in a model's overall parameters. This validated our guess that changes in the last layer's parameters can effectively represent changes across all parameters of a model

### 5.4 Performance on Different Datasets

Tables 1, 2, 3, and 4 present the classification accuracies on different datasets with different methods for various sizes of the unlearning

**Table 1: Means and standard deviations (percentage) of classification accuracy on CIFAR10.**

| | Method | $|D_f|$=16 | $|D_f|$=32 | $|D_f|$=64 | $|D_f|$=128 | $|D_f|$=256 | $|D_f|$=512 |
|---|---|---|---|---|---|---|---|
| $D_f$ | *BEFORE* | 97.92 ± 4.93 | 98.12 ± 3.39 | 98.33 ± 3.15 | 98.39 ± 2.73 | 98.26 ± 2.79 | 97.83 ± 3.17 |
| | NegGrad | 16.25 ± 33.68 | 3.33 ± 4.90 | 2.92 ± 4.30 | 3.12 ± 4.60 | 3.62 ± 5.19 | 3.48 ± 4.98 |
| | RandL | 0.00 ± 0.00 | 0.00 ± 0.00 | 0.00 ± 0.00 | 0.00 ± 0.00 | 0.00 ± 0.00 | 0.00 ± 0.00 |
| | Bad Teacher | 11.25 ± 8.29 | 9.38 ± 4.70 | 11.46 ± 3.68 | 12.34 ± 4.64 | 13.18 ± 4.34 | 16.07 ± 7.20 |
| | Amnesiac | 0.00 ± 0.00 | 0.00 ± 0.00 | 0.00 ± 0.00 | 0.00 ± 0.00 | 0.00 ± 0.00 | 0.00 ± 0.00 |
| | $\delta$-Targeted[1] | 0.00 ± 0.00 | 0.00 ± 0.00 | 0.00 ± 0.00 | 0.00 ± 0.00 | 0.00 ± 0.00 | 0.00 ± 0.00 |
| | $\delta$-Targeted[3] | 0.00 ± 0.00 | 0.00 ± 0.00 | 0.00 ± 0.00 | 0.00 ± 0.00 | 0.18 ± 0.35 | 0.90 ± 1.40 |
| $D_r$ | *BEFORE* | 97.88 ± 3.00 | 97.88 ± 3.00 | 97.88 ± 3.00 | 97.88 ± 3.00 | 97.87 ± 3.00 | 97.88 ± 2.99 |
| | NegGrad | 69.62 ± 22.88 | 53.10 ± 24.18 | 32.81 ± 15.36 | 17.46 ± 6.98 | 9.29 ± 1.36 | 6.64 ± 2.49 |
| | RandL | 87.98 ± 6.54 | 84.92 ± 7.67 | 74.79 ± 10.21 | 59.97 ± 9.95 | 42.47 ± 9.91 | 28.87 ± 7.34 |
| | Bad Teacher | 78.09 ± 6.95 | 78.11 ± 7.25 | 75.75 ± 7.26 | 69.23 ± 10.53 | 56.55 ± 14.55 | 40.50 ± 13.56 |
| | Amnesiac | 76.83 ± 4.44 | 75.84 ± 4.27 | 73.77 ± 3.97 | 68.89 ± 4.39 | 54.21 ± 7.68 | 34.29 ± 6.20 |
| | $\delta$-Targeted[1] | **94.35 ± 6.48** | **93.73 ± 7.13** | 89.34 ± 10.51 | 84.57 ± 10.22 | 75.32 ± 7.38 | 57.13 ± 2.69 |
| | $\delta$-Targeted[3] | 93.24 ± 4.51 | 93.61 ± 5.50 | **91.98 ± 7.80** | **90.55 ± 8.50** | **86.22 ± 7.98** | **73.78 ± 4.74** |
| $D_t$ | *BEFORE* | 89.62 ± 5.62 | 89.62 ± 5.62 | 89.62 ± 5.62 | 89.62 ± 5.62 | 89.62 ± 5.62 | 89.62 ± 5.62 |
| | NegGrad | 64.41 ± 20.62 | 48.77 ± 21.49 | 29.71 ± 13.12 | 15.61 ± 5.62 | 8.70 ± 1.45 | 6.39 ± 2.58 |
| | RandL | 80.15 ± 7.11 | 76.86 ± 7.97 | 67.25 ± 9.89 | 53.41 ± 8.61 | 37.44 ± 8.49 | 25.10 ± 6.12 |
| | Bad Teacher | 73.53 ± 4.35 | 73.29 ± 5.23 | 71.26 ± 5.47 | 65.24 ± 9.19 | 53.46 ± 13.09 | 38.69 ± 12.63 |
| | Amnesiac | 72.14 ± 2.42 | 71.14 ± 3.34 | 69.16 ± 3.98 | 64.62 ± 4.46 | 50.70 ± 6.62 | 32.24 ± 5.38 |
| | $\delta$-Targeted[1] | **85.73 ± 7.47** | **85.26 ± 8.09** | 81.17 ± 10.64 | 76.43 ± 10.00 | 67.36 ± 7.11 | 50.68 ± 2.56 |
| | $\delta$-Targeted[3] | 84.59 ± 5.74 | 85.08 ± 6.58 | **83.62 ± 8.55** | **82.17 ± 9.07** | **77.75 ± 8.32** | **65.93 ± 4.64** |

**Table 2: Means and standard deviations (percentage) of classification accuracy on CIFAR100.**

| | Method | $|D_f|$=16 | $|D_f|$=32 | $|D_f|$=64 | $|D_f|$=128 | $|D_f|$=256 | $|D_f|$=512 |
|---|---|---|---|---|---|---|---|
| $D_f$ | *BEFORE* | 95.42 ± 8.06 | 93.96 ± 9.98 | 94.48 ± 7.94 | 94.84 ± 7.19 | 94.71 ± 7.38 | 94.92 ± 7.17 |
| | NegGrad | 0.42 ± 1.56 | 0.21 ± 0.78 | 0.21 ± 0.53 | 7.03 ± 24.85 | 0.26 ± 0.47 | 0.23 ± 0.43 |
| | RandL | 0.00 ± 0.00 | 0.00 ± 0.00 | 0.00 ± 0.00 | 0.00 ± 0.00 | 0.00 ± 0.00 | 0.00 ± 0.00 |
| | Bad Teacher | 1.25 ± 3.39 | 1.25 ± 1.91 | 2.08 ± 2.03 | 2.24 ± 2.23 | 3.12 ± 2.58 | 4.21 ± 4.37 |
| | Amnesiac | 0.00 ± 0.00 | 0.00 ± 0.00 | 0.00 ± 0.00 | 0.00 ± 0.00 | 0.00 ± 0.00 | 0.00 ± 0.00 |
| | $\delta$-Targeted[1] | 0.00 ± 0.00 | 0.00 ± 0.00 | 0.00 ± 0.00 | 0.00 ± 0.00 | 0.00 ± 0.00 | 0.00 ± 0.00 |
| | $\delta$-Targeted[3] | 0.00 ± 0.00 | 0.00 ± 0.00 | 0.00 ± 0.00 | 0.00 ± 0.00 | 0.00 ± 0.00 | 0.00 ± 0.00 |
| $D_r$ | *BEFORE* | 95.04 ± 6.97 | 95.04 ± 6.97 | 95.04 ± 6.96 | 95.04 ± 6.97 | 95.04 ± 6.97 | 95.04 ± 6.97 |
| | NegGrad | 62.90 ± 19.23 | 59.81 ± 25.59 | 53.14 ± 29.77 | 47.59 ± 34.69 | 41.18 ± 29.50 | 33.36 ± 23.49 |
| | RandL | 82.07 ± 11.83 | 83.30 ± 14.32 | 81.64 ± 17.73 | 78.32 ± 22.15 | 73.81 ± 26.68 | 67.00 ± 31.73 |
| | Bad Teacher | 44.56 ± 21.53 | 43.94 ± 20.06 | 40.29 ± 17.11 | 31.14 ± 17.08 | 23.36 ± 14.10 | 15.56 ± 9.77 |
| | Amnesiac | 41.52 ± 16.65 | 40.75 ± 15.86 | 38.51 ± 14.30 | 35.55 ± 13.66 | 26.92 ± 14.24 | 16.78 ± 10.37 |
| | $\delta$-Targeted[1] | **93.39 ± 6.96** | **92.85 ± 7.43** | **92.00 ± 8.63** | **90.64 ± 10.10** | 89.32 ± 11.06 | 86.20 ± 11.15 |
| | $\delta$-Targeted[3] | 90.88 ± 5.42 | 91.58 ± 6.31 | 91.09 ± 7.33 | 90.28 ± 8.20 | **89.70 ± 8.62** | **88.39 ± 9.12** |
| $D_t$ | *BEFORE* | 67.07 ± 8.89 | 67.07 ± 8.89 | 67.07 ± 8.89 | 67.07 ± 8.89 | 67.07 ± 8.89 | 67.07 ± 8.89 |
| | NegGrad | 43.66 ± 11.78 | 41.15 ± 15.74 | 36.29 ± 18.85 | 32.19 ± 23.34 | 27.50 ± 19.32 | 22.38 ± 15.39 |
| | RandL | 55.43 ± 7.83 | 56.48 ± 9.79 | 55.72 ± 12.13 | 53.40 ± 14.77 | 49.57 ± 17.30 | 43.51 ± 19.77 |
| | Bad Teacher | 34.26 ± 12.22 | 33.89 ± 11.57 | 31.59 ± 10.02 | 24.47 ± 11.97 | 18.69 ± 10.84 | 12.91 ± 8.09 |
| | Amnesiac | 32.08 ± 8.73 | 31.51 ± 8.30 | 30.01 ± 7.45 | 27.75 ± 7.52 | 20.64 ± 8.64 | 13.03 ± 6.84 |
| | $\delta$-Targeted[1] | **63.24 ± 7.11** | **63.08 ± 6.99** | **62.87 ± 7.98** | **62.13 ± 8.73** | 60.85 ± 9.22 | 57.84 ± 8.43 |
| | $\delta$-Targeted[3] | 60.70 ± 5.37 | 61.68 ± 5.97 | 61.92 ± 6.97 | 61.67 ± 7.41 | **60.97 ± 7.67** | **59.43 ± 7.62** |

sample $D_f$. Note that the results in these tables are obtained by averaging over different neural network architectures, including ResNet-18, ResNet-50, and ViT. The results for individual neural network architectures can be found in App. B.

We first examine the performance on $D_f$, which is the subset of data we want our model to unlearn. The tables show that all methods can effectively unlearn the data, resulting in near-zero accuracy on $D_f$. Secondly, looking at the performance on $D_r$, which is the remaining examples that the model should still remember, it is evident that both $\delta$-Targeted[1] and $\delta$-Targeted[3] consistently outperform the other methods. In particular, $\delta$-Targeted[3] demonstrates superior

performance compared to other methods, as it mostly achieves the highest accuracy across different sizes of $D_f$, indicating that it effectively retains the model's ability to predict the remaining sample despite the unlearning process.

Lastly, on $D_t$, the test dataset, both $\delta$-Targeted[1] and $\delta$-Targeted[3] again demonstrate superior performance. Despite the unlearning process, these methods ensure the model maintains its overall predictive performance on unseen data, outperforming other methods across different sizes of $D_f$.

When the size of the unlearning sample $D_f$ is large, $\delta$-Targeted[3] consistently outperforms the other methods. It effectively unlearns

**Table 3: Means and standard deviations (percentage) of classification accuracy on Fashion-MNIST.**

| | Method | $|D_f|$=16 | $|D_f|$=32 | $|D_f|$=64 | $|D_f|$=128 | $|D_f|$=256 | $|D_f|$=512 |
|---|---|---|---|---|---|---|---|
| | *BEFORE* | 96.25 ± 5.00 | 95.42 ± 4.25 | 94.79 ± 3.16 | 94.95 ± 2.73 | 95.13 ± 2.11 | 95.76 ± 1.78 |
| | NegGrad | 2.92 ± 6.80 | 2.71 ± 6.54 | 3.65 ± 6.42 | 2.24 ± 4.33 | 3.36 ± 5.03 | 3.48 ± 4.99 |
| | RandL | 0.00 ± 0.00 | 0.00 ± 0.00 | 0.00 ± 0.00 | 0.00 ± 0.00 | 0.00 ± 0.00 | 0.46 ± 0.68 |
| $D_f$ | Bad Teacher | 7.92 ± 8.06 | 6.04 ± 5.41 | 6.77 ± 5.84 | 6.93 ± 5.51 | 8.83 ± 6.67 | 8.87 ± 7.05 |
| | Amnesiac | 0.00 ± 0.00 | 0.42 ± 1.56 | 2.50 ± 3.41 | 6.30 ± 8.82 | 12.37 ± 13.05 | 10.68 ± 10.84 |
| | $\delta$-Targeted[1] | 0.00 ± 0.00 | 0.00 ± 0.00 | 0.62 ± 1.11 | 1.09 ± 1.91 | 4.56 ± 6.83 | 17.50 ± 24.87 |
| | $\delta$-Targeted[3] | 11.25 ± 16.65 | 10.42 ± 15.21 | 15.62 ± 22.37 | 22.40 ± 31.67 | 25.55 ± 36.15 | 25.85 ± 36.57 |
| | *BEFORE* | 96.04 ± 1.37 | 96.04 ± 1.37 | 96.04 ± 1.37 | 96.04 ± 1.37 | 96.04 ± 1.36 | 96.04 ± 1.36 |
| | NegGrad | 53.98 ± 24.96 | 41.26 ± 22.91 | 30.46 ± 14.70 | 17.86 ± 7.91 | 11.69 ± 3.20 | 8.02 ± 2.19 |
| | RandL | 72.05 ± 17.74 | 64.08 ± 23.78 | 53.59 ± 24.06 | 40.22 ± 19.96 | 27.35 ± 14.38 | 18.87 ± 8.96 |
| $D_r$ | Bad Teacher | 79.16 ± 10.05 | 76.20 ± 12.20 | 71.15 ± 15.48 | 63.17 ± 17.87 | 49.02 ± 17.18 | 37.71 ± 14.54 |
| | Amnesiac | 81.51 ± 6.08 | 80.15 ± 5.16 | 74.83 ± 7.61 | 69.01 ± 8.52 | 56.34 ± 6.37 | 40.00 ± 3.08 |
| | $\delta$-Targeted[1] | 91.76 ± 3.26 | 88.56 ± 3.31 | 82.73 ± 3.38 | 72.52 ± 3.96 | 57.43 ± 5.38 | 45.50 ± 11.91 |
| | $\delta$-Targeted[3] | **92.47 ± 1.68** | **91.98 ± 2.15** | **90.04 ± 2.48** | **85.04 ± 2.44** | **75.70 ± 4.92** | **61.69 ± 11.73** |
| | *BEFORE* | 91.95 ± 0.33 | 91.95 ± 0.33 | 91.95 ± 0.33 | 91.95 ± 0.33 | 91.95 ± 0.33 | 91.95 ± 0.33 |
| | NegGrad | 51.19 ± 23.07 | 38.70 ± 21.18 | 28.15 ± 13.11 | 16.12 ± 6.77 | 10.58 ± 2.58 | 7.39 ± 2.37 |
| | RandL | 68.66 ± 16.13 | 60.82 ± 21.88 | 50.34 ± 22.10 | 37.26 ± 18.06 | 24.65 ± 12.58 | 16.47 ± 7.52 |
| $D_t$ | Bad Teacher | 77.94 ± 10.06 | 74.83 ± 12.05 | 69.89 ± 15.42 | 61.94 ± 17.84 | 47.99 ± 17.12 | 37.14 ± 14.48 |
| | Amnesiac | 80.24 ± 6.23 | 78.87 ± 5.31 | 73.62 ± 7.83 | 67.77 ± 8.67 | 55.18 ± 6.86 | 38.94 ± 3.54 |
| | $\delta$-Targeted[1] | 87.99 ± 1.62 | 84.85 ± 2.47 | 79.04 ± 3.16 | 68.78 ± 4.71 | 54.04 ± 6.72 | 42.59 ± 13.57 |
| | $\delta$-Targeted[3] | **88.89 ± 0.85** | **88.34 ± 0.75** | **86.27 ± 1.27** | **81.36 ± 2.64** | **72.14 ± 6.60** | **58.54 ± 13.48** |

**Table 4: Means and standard deviations (percentage) of classification accuracy on Tiny-ImageNet.**

| | Method | $|D_f|$=16 | $|D_f|$=32 | $|D_f|$=64 | $|D_f|$=128 | $|D_f|$=256 | $|D_f|$=512 |
|---|---|---|---|---|---|---|---|
| | *BEFORE* | 64.53 ± 24.87 | 65.17 ± 24.07 | 64.59 ± 23.62 | 65.13 ± 23.10 | 65.32 ± 22.02 | 66.52 ± 23.84 |
| | NegGrad | 0.00 ± 0.00 | 0.00 ± 0.00 | 0.00 ± 0.00 | 0.00 ± 0.00 | 0.00 ± 0.00 | 0.00 ± 0.00 |
| | RandL | 0.00 ± 0.00 | 0.00 ± 0.00 | 0.00 ± 0.00 | 0.00 ± 0.00 | 0.00 ± 0.00 | 0.00 ± 0.00 |
| $D_f$ | Bad Teacher | 0.42 ± 1.56 | 0.21 ± 0.78 | 0.00 ± 0.00 | 0.42 ± 0.48 | 0.31 ± 0.36 | 0.42 ± 0.40 |
| | Amnesiac | 0.00 ± 0.00 | 0.00 ± 0.00 | 0.00 ± 0.00 | 0.00 ± 0.00 | 0.00 ± 0.00 | 0.00 ± 0.00 |
| | $\delta$-Targeted[1] | 0.00 ± 0.00 | 0.00 ± 0.00 | 0.00 ± 0.00 | 0.00 ± 0.00 | 0.00 ± 0.00 | 0.00 ± 0.00 |
| | $\delta$-Targeted[3] | 0.00 ± 0.00 | 0.00 ± 0.00 | 0.00 ± 0.00 | 0.00 ± 0.00 | 0.00 ± 0.00 | 0.01 ± 0.05 |
| | *BEFORE* | 71.22 ± 15.44 | 71.22 ± 15.44 | 71.22 ± 15.44 | 71.22 ± 15.44 | 71.22 ± 15.44 | 73.26 ± 15.19 |
| | NegGrad | 44.78 ± 9.53 | 48.91 ± 9.60 | 49.47 ± 10.10 | 43.80 ± 11.20 | 38.23 ± 11.02 | 34.23 ± 8.57 |
| | RandL | 46.29 ± 11.93 | 47.89 ± 15.37 | 50.41 ± 17.50 | 52.53 ± 17.65 | 54.07 ± 15.93 | 53.48 ± 14.17 |
| $D_r$ | Bad Teacher | 25.22 ± 4.70 | 25.55 ± 5.01 | 24.58 ± 3.91 | 23.25 ± 5.12 | 19.22 ± 5.42 | 16.43 ± 6.06 |
| | Amnesiac | 23.86 ± 8.20 | 23.34 ± 7.86 | 23.03 ± 7.56 | 22.90 ± 8.15 | 22.51 ± 7.88 | 19.23 ± 8.16 |
| | $\delta$-Targeted[1] | **61.68 ± 15.20** | **61.02 ± 16.13** | 59.60 ± 17.15 | 59.19 ± 17.17 | 59.62 ± 16.62 | 57.34 ± 17.32 |
| | $\delta$-Targeted[3] | 60.84 ± 12.44 | 59.67 ± 13.10 | **60.13 ± 14.65** | **60.05 ± 15.72** | **60.04 ± 16.49** | **59.98 ± 16.08** |
| | *BEFORE* | 52.51 ± 0.23 | 52.51 ± 0.23 | 52.51 ± 0.23 | 52.51 ± 0.23 | 52.51 ± 0.23 | 52.53 ± 0.24 |
| | NegGrad | 39.32 ± 2.95 | 40.42 ± 2.47 | 40.13 ± 2.42 | 38.30 ± 2.62 | 32.08 ± 3.72 | 30.09 ± 2.27 |
| | RandL | 40.37 ± 3.05 | 40.58 ± 4.67 | 40.88 ± 5.49 | 42.48 ± 4.41 | 43.67 ± 3.74 | 44.17 ± 3.28 |
| $D_t$ | Bad Teacher | 24.66 ± 5.75 | 25.35 ± 5.11 | 26.43 ± 6.38 | 22.85 ± 5.12 | 18.95 ± 5.66 | 16.92 ± 6.53 |
| | Amnesiac | 22.98 ± 7.84 | 24.58 ± 9.00 | 23.35 ± 8.36 | 24.21 ± 9.51 | 23.13 ± 9.24 | 17.96 ± 8.23 |
| | $\delta$-Targeted[1] | **51.28 ± 1.30** | **50.68 ± 2.18** | 49.79 ± 4.38 | 48.59 ± 2.98 | 47.97 ± 3.28 | 46.90 ± 3.68 |
| | $\delta$-Targeted[3] | 50.04 ± 2.00 | 50.39 ± 1.17 | **49.84 ± 3.36** | **50.77 ± 2.69** | **49.37 ± 2.29** | **47.02 ± 3.74** |

harder-to-forget examples by emphasizing the importance of harder-to-forget examples in the unlearning set. This makes it particularly advantageous for larger unlearning sets. However, when the unlearning set is small, $\delta$-Targeted[1] tends to outperform $\delta$-Targeted[3]. This might be because when all examples can be easily unlearned, placing extra emphasis on certain examples (as in $\lambda = 3$) does not provide additional benefits. A potential reason could be that, when the to-be-forgotten sample size is small, these examples can be unlearned in just a few steps. Emphasizing the loss too much at the beginning can lead to large parameter updates, which adversely affect the model's performance on the remaining data.

## 6 Conclusion

In this work, we presented $\delta$-Targeted$^\gamma$, a method designed to retain model performance after removing a specific training sample without the need for reassessing the remaining sample. Theoretically, we demonstrated that our method results in more concentrated parameter updates than those introduced by gradient ascent, thereby effectively reducing the performance drops commonly observed after unlearning. Our empirical evaluations further corroborate these findings, as $\delta$-Targeted$^\gamma$ consistently achieves state-of-the-art results across a variety of benchmark image datasets, underscoring both its theoretical soundness and practical efficacy.

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

## A Proof of Theorem 4.1

We define some notations used in the proof first. Let $(x, y)$ by an example. Let $o$ represent the activation just before the last layer of the pre-trained model. For the label $y$, its corresponding one-hot encoding is denoted by $y$. The estimated class posteriors $p = \sigma \circ h_{\hat{\phi}}(o)$ are defined by the output of the softmax function applied to the linear transformation. It can be further expanded to $p = \sigma(\hat{\phi}o) = \sigma(z)$, where $\hat{\phi} \in \mathbb{R}^{C \times M}$ represents the parameter of the last fully connected layer, $z = \hat{\phi}o$ represents the input to the softmax function, and $\sigma$ is the softmax function.

Firstly, we derive the gradient of cross-entropy loss with respect to the last layer parameter $\phi$. The cross-entropy loss function $\ell(p, y)$ is defined as:

$$\ell(p, y) = -\sum_{k=1}^{C} y_k \cdot \log(p_k). \tag{3}$$

The gradient of $\ell$ with respect to the predicted class-posterior probability $p_k = P(Y = k | X = x)$ can be calculated as:

$$\frac{\partial \ell}{\partial p_k} = -\frac{y_k}{p_k}. \tag{4}$$

Consider applying softmax function $\sigma(z)_i$ to calculate $i$-th class posterior $p_i = \sigma(z)_i = \frac{e^{z_i}}{\sum_{j=1}^{C} e^{z_j}}$.

Case when $i = k$, the derivative $\frac{\partial p_i}{\partial z_k} = \frac{\partial p_k}{\partial z_k}$ is

$$\frac{\partial p_k}{\partial z_k} = \frac{\sum_{j=1}^{C} e^{z_j} \cdot e^{z_k} - e^{z_k} \cdot e^{z_k}}{(\sum_{j=1}^{C} e^{z_j})^2}$$

$$= \frac{e^{z_k}}{\sum_{j=1}^{C} e^{z_j}} \cdot (1 - \frac{e^{z_k}}{\sum_{j=1}^{C} e^{z_j}})$$

$$= p_k(1 - p_k).$$

Case when $i \neq k$, the derivative $\frac{\partial p_i}{\partial z_k}$ is

$$\frac{\partial p_i}{\partial z_k} = \frac{0 - e^{z_k} \cdot e^{z_i}}{(\sum_{j=1}^{C} e^{z_j})^2}$$

$$= -\frac{e^{z_k} \cdot e^{z_i}}{(\sum_{j=1}^{C} e^{z_j})^2}$$

$$= -p_k p_i.$$

Applying the chain rule, the gradient of $\ell$ with respect to $z_k$ becomes:

$$\frac{\partial \ell}{\partial z_k} = \sum_i \frac{\partial \ell}{\partial p_i} \frac{\partial p_i}{\partial z_k} = -y_k(1 - p_k) + \sum_{j \neq k} y_j p_k. \tag{5}$$

Note that $y$ is the one-hot encoding form of $y$, $y_k$ can only include a value of either 0 or 1. The above equation simplifies to $p_k - 1$ when $y = k$ and $p_k$ when $y \neq k$. In matrix form, we can rewrite it as:

$$\frac{\partial \ell}{\partial z} = p - y. \tag{6}$$

Furthermore, the derivative of logits $z$ with respect to the last-layer parameter $\phi$ is $\frac{\partial z}{\partial \phi} = o^T$. Then, by applying the chain rule again, the gradient of the loss $\ell$ with respect to the parameter $\phi$ is given by:

$$\frac{\partial \ell}{\partial \phi} = \frac{\partial \ell}{\partial z} \frac{\partial z}{\partial \phi} = (p - y)o^T. \tag{7}$$

To unlearn the example $(x, y)$, we update $\phi$ to $\phi'$ using gradient ascent: $\phi' = \phi + \eta' \nabla_{\hat{\phi}} \ell(h_{\hat{\phi}}(o), y)$, where $\eta' \in R$ is the learning rate, and $\eta' > 0$. The updated elements of $\phi'$ are defined as:

$$\phi'_y = \phi_y + \eta'(p_y - 1)o^T,$$

$$\phi'_k = \phi_k + \eta'(p_k)o^T, \text{ for } k \neq y.$$

let $\Delta \boldsymbol{p} = \boldsymbol{p} - \boldsymbol{y}$, we write the above equations by using gradient ascent in matrix form

$$\phi' = \phi + \eta'(\boldsymbol{p} - \boldsymbol{y})\boldsymbol{o}^T$$
$$= \phi + \eta'\Delta\boldsymbol{p}\boldsymbol{o}^T.$$

We also consider an update of $\phi$ to $\phi''$ using gradient descent: $\phi'' = \phi - \eta\nabla_{\hat{\phi}}L(h_{\hat{\phi}}(\boldsymbol{o}), y')$, where $\eta \in R$ is the learning rate, and $\eta > 0$. The updated elements of $\phi''$ are defined as:

$$\phi''_{y'} = \phi_{y'} + \eta(1 - \boldsymbol{p}_{y'})\boldsymbol{o}^T,$$
$$\phi''_k = \phi_k + \eta\boldsymbol{p}_k\boldsymbol{o}^T,$$
$$\phi''_y = \phi_y + \eta\boldsymbol{p}_y\boldsymbol{o}^T, \text{ for } k \neq y, y'.$$

Let $\Delta \boldsymbol{p}' = \boldsymbol{y}' - \boldsymbol{p}$, we write the above equations by using gradient descent in matrix form

$$\phi'' = \phi + \eta(\boldsymbol{y}' - \boldsymbol{p})\boldsymbol{o}^T$$
$$= \phi + \eta\Delta\boldsymbol{p}'\boldsymbol{o}^T.$$

The $L_1$-norms of $\Delta\phi'$ and $\Delta\phi''$ are given by:

$$||\Delta\phi'||_1 = \eta'||\Delta\boldsymbol{p}\boldsymbol{o}^T||_1,$$
$$||\Delta\phi''||_1 = \eta||\Delta\boldsymbol{p}'\boldsymbol{o}^T||_1.$$

We now compare the effect of gradient ascent and the gradient descent when $\Delta\phi'$ and $\Delta\phi''$ have the same $L_1$ norm. Recall $\Delta\boldsymbol{p} = \boldsymbol{p} - \boldsymbol{y}$ and $\Delta\boldsymbol{p}' = \boldsymbol{y}' - \boldsymbol{p}$, we can derive the following points:

- For $i \neq y$ and $i \neq y'$, we have $|\Delta\boldsymbol{p}'_i| = |-\boldsymbol{p}_i|$ and $|\Delta\boldsymbol{p}_i| = |\boldsymbol{p}_i|$.
- When $i = y$, we have $|\Delta\boldsymbol{p}'_i| = |-\boldsymbol{p}_y|$ and $|\Delta\boldsymbol{p}_i| = |\boldsymbol{p}_y - 1|$.
- When $i = y'$, we have $|\Delta\boldsymbol{p}'_i| = |1 - \boldsymbol{p}_{y'}|$ and $|\Delta\boldsymbol{p}_i| = |-\boldsymbol{p}_{y'}|$.

Therefore, the $L_1$-norms of $\Delta\boldsymbol{p}$ and $\Delta\boldsymbol{p}'$ are:

$$||\Delta\boldsymbol{p}||_1 = |\boldsymbol{p}_{y'}| + |1 - \boldsymbol{p}_y| + \sum_{i \neq y, y'} |\boldsymbol{p}_i| = 2 - 2\boldsymbol{p}_y,$$

$$||\Delta\boldsymbol{p}'||_1 = |1 - \boldsymbol{p}_{y'}| + |\boldsymbol{p}_y| + \sum_{i \neq y, y'} |-\boldsymbol{p}_i| = 2 - 2\boldsymbol{p}_{y'}.$$

Given the fact that $y$ is most confident label, then $\boldsymbol{p}_y > \boldsymbol{p}_{y'}$, we then have:

$$||\Delta\boldsymbol{p}||_1 - ||\Delta\boldsymbol{p}'||_1 = 2 - 2\boldsymbol{p}_y - (2 - 2\boldsymbol{p}_{y'})$$
$$= -2\boldsymbol{p}_y + 2\boldsymbol{p}_{y'} < 0.$$

Let $\eta' = \frac{1}{2 - 2\boldsymbol{p}_y}$ and $\eta = \frac{1}{2 - 2\boldsymbol{p}_{y'}}$ to ensure $\Delta\phi'$ and $\Delta\phi''$ have the same $L_1$ norm. Then, we can rewrite the update rules for $\phi'$ and $\phi''$ as follows:

$$\phi'_{*,j} = \phi_{*,j} + \eta'(\boldsymbol{p} - \boldsymbol{y})\boldsymbol{o}_j = \phi_{*,j} + \frac{1}{2 - 2\boldsymbol{p}_y}(\boldsymbol{p} - \boldsymbol{y})\boldsymbol{o}_j,$$

$$\phi''_{*,j} = \phi_{*,j} + \eta(\boldsymbol{y}' - \boldsymbol{p})\boldsymbol{o}_j = \phi_{*,j} + \frac{1}{2 - 2\boldsymbol{p}_{y'}}(\boldsymbol{y}' - \boldsymbol{p})\boldsymbol{o}_j,$$

where $\boldsymbol{M}_{*,j}$ denote the $j$-th column in a matrix $\boldsymbol{M}$.

Now let us discuss the following two cases to compare the change of parameters:

1). For all $i \neq y, y'$, we have:

$$\phi'_{ij} = \phi_{ij} + \frac{1}{2 - 2\boldsymbol{p}_y}\boldsymbol{p}_i\boldsymbol{o}_j, \tag{8}$$

$$\phi''_{ij} = \phi_{ij} - \frac{1}{2 - 2\boldsymbol{p}_{y'}}\boldsymbol{p}_i\boldsymbol{o}_j. \tag{9}$$

Given that $\boldsymbol{p}_{y'} < \boldsymbol{p}_y$, it follows that $|\phi''_{ij}| - |\phi'_{ij}| < 0$. Thus, when the $L_1$ norm of $\Delta\phi'$ and $\Delta\phi''$ are the same, the gradient ascent method leads to more changes in the parameters for predicting $\boldsymbol{p}_i$ for $i \neq y$ and $i \neq y'$ compared to gradient descent.

2). For $i = y$, we have:

$$\phi'_{ij} = \phi_{ij} + \frac{1}{2 - 2p_y}(p_y - 1)o_j, \qquad (10)$$

$$\phi''_{ij} = \phi_{ij} - \frac{1}{2 - 2p_{y'}}p_y o_j. \qquad (11)$$

Observing that $|\phi''_{ij}| - |\phi'_{ij}| = \frac{-1 + p_{y'} + p_y}{2(1 - p_{y'})} < 0$ when having more than two classes. Then, when the $L_1$ norms of $\Delta\phi'$ and $\Delta\phi''$ are identical, the gradient ascent method leads to more changes in the parameters for predicting $p_y$ than gradient descent.

Combining the findings for both cases, $|\phi''_{ij}| - |\phi'_{ij}| < 0$. As $L_1$ norms of $\Delta\phi'$ and $\Delta\phi''$ are identical, we conclude that only for $i = y'$, the value of $|\phi''_{ij}| - |\phi'_{ij}| > 0$. This means our method only leads to large changes in the parameters for predicting pseudo-label $y'$.

When more than two classes, our method results in parameter updates that are more concentrated on the weights associated with the pseudo-label $y'$, while gradient ascent distributes updates more uniformly across all classes, which completes the proof.

## B  Additional Experiment Results

In this section, we provide additional experiment results, including performance on different datasets with various neural network structures, changes in last-layer parameters for different sample unlearning methods, and changes in all parameters for different sample unlearning methods. We also explore the relationships between changes in last-layer parameters and changes in all parameters, as well as the relationships between a model's accuracy on the remaining examples and changes in all parameters.

### B.1  Performance on CIFAR10 with Different Neural Network Structures

Table 5: Means and standard deviations (percentage) of classification accuracy on CIFAR10 with ResNet-18.

| | Method | $|D_f|$=16 | $|D_f|$=32 | $|D_f|$=64 | $|D_f|$=128 | $|D_f|$=256 | $|D_f|$=512 |
|---|---|---|---|---|---|---|---|
| $D_f$ | *BEFORE* | 100.00 ± 0.00 | 100.00 ± 0.00 | 100.00 ± 0.00 | 100.00 ± 0.00 | 100.00 ± 0.00 | 100.00 ± 0.00 |
| | NegGrad | 40.00 ± 48.99 | 0.00 ± 0.00 | 0.00 ± 0.00 | 0.00 ± 0.00 | 0.00 ± 0.00 | 0.00 ± 0.00 |
| | RandL | 0.00 ± 0.00 | 0.00 ± 0.00 | 0.00 ± 0.00 | 0.00 ± 0.00 | 0.00 ± 0.00 | 0.00 ± 0.00 |
| | Bad Teacher | 7.50 ± 7.29 | 10.62 ± 4.24 | 11.56 ± 2.90 | 9.53 ± 2.39 | 10.23 ± 1.72 | 11.48 ± 1.18 |
| | Amnesiac | 0.00 ± 0.00 | 0.00 ± 0.00 | 0.00 ± 0.00 | 0.00 ± 0.00 | 0.00 ± 0.00 | 0.00 ± 0.00 |
| | $\delta$-Targeted[1] | 0.00 ± 0.00 | 0.00 ± 0.00 | 0.00 ± 0.00 | 0.00 ± 0.00 | 0.00 ± 0.00 | 0.00 ± 0.00 |
| | $\delta$-Targeted[3] | 0.00 ± 0.00 | 0.00 ± 0.00 | 0.00 ± 0.00 | 0.00 ± 0.00 | 0.00 ± 0.00 | 0.00 ± 0.00 |
| $D_r$ | *BEFORE* | 99.99 ± 0.00 | 99.99 ± 0.00 | 99.99 ± 0.00 | 99.99 ± 0.00 | 99.99 ± 0.00 | 99.99 ± 0.00 |
| | NegGrad | 92.52 ± 7.43 | 74.05 ± 10.62 | 46.34 ± 7.70 | 25.25 ± 3.48 | 9.28 ± 1.58 | 4.97 ± 0.84 |
| | RandL | 92.17 ± 3.04 | 90.96 ± 1.34 | 83.47 ± 4.08 | 67.99 ± 3.01 | 50.06 ± 2.85 | 34.35 ± 1.25 |
| | Bad Teacher | 77.73 ± 4.66 | 81.09 ± 1.33 | 81.18 ± 1.27 | 77.40 ± 1.62 | 67.22 ± 4.14 | 45.04 ± 6.04 |
| | Amnesiac | 78.64 ± 1.34 | 79.30 ± 1.26 | 78.10 ± 1.46 | 74.23 ± 1.23 | 60.68 ± 1.34 | 35.88 ± 2.96 |
| | $\delta$-Targeted[1] | **98.39 ± 1.05** | **98.83 ± 0.59** | 96.61 ± 1.88 | 92.33 ± 1.88 | 79.59 ± 2.51 | 56.98 ± 2.33 |
| | $\delta$-Targeted[3] | 96.19 ± 1.65 | 97.91 ± 0.53 | **97.57 ± 0.55** | **96.63 ± 0.62** | **91.43 ± 0.93** | **78.17 ± 0.89** |
| $D_t$ | *BEFORE* | 93.13 ± 0.00 | 93.13 ± 0.00 | 93.13 ± 0.00 | 93.13 ± 0.00 | 93.13 ± 0.00 | 93.13 ± 0.00 |
| | NegGrad | 84.82 ± 7.64 | 66.76 ± 9.92 | 40.66 ± 6.60 | 21.56 ± 3.65 | 8.08 ± 1.33 | 4.72 ± 0.84 |
| | RandL | 84.33 ± 2.72 | 82.71 ± 1.18 | 75.24 ± 4.03 | 59.91 ± 2.60 | 43.35 ± 2.96 | 28.88 ± 1.21 |
| | Bad Teacher | 74.16 ± 3.86 | 76.79 ± 1.39 | 77.12 ± 0.87 | 74.00 ± 1.40 | 64.28 ± 4.14 | 43.15 ± 5.49 |
| | Amnesiac | 74.77 ± 1.19 | 75.50 ± 1.03 | 74.40 ± 1.48 | 70.59 ± 1.00 | 57.57 ± 1.35 | 34.47 ± 2.83 |
| | $\delta$-Targeted[1] | **90.17 ± 1.21** | **90.72 ± 0.75** | 88.19 ± 2.00 | 83.61 ± 2.05 | 71.06 ± 2.31 | 50.03 ± 1.96 |
| | $\delta$-Targeted[3] | 88.06 ± 1.55 | 89.64 ± 0.74 | **89.30 ± 0.63** | **88.17 ± 0.62** | **82.68 ± 1.00** | **69.63 ± 1.05** |

**Table 6: Means and standard deviations (percentage) of classification accuracy on CIFAR10 with ResNet-50.**

|  | Method | $|D_f|$=16 | $|D_f|$=32 | $|D_f|$=64 | $|D_f|$=128 | $|D_f|$=256 | $|D_f|$=512 |
|---|---|---|---|---|---|---|---|
| $D_f$ | *BEFORE* | 100.00 ± 0.00 | 100.00 ± 0.00 | 100.00 ± 0.00 | 100.00 ± 0.00 | 100.00 ± 0.00 | 100.00 ± 0.00 |
|  | NegGrad | 0.00 ± 0.00 | 0.00 ± 0.00 | 0.00 ± 0.00 | 0.00 ± 0.00 | 0.00 ± 0.00 | 0.00 ± 0.00 |
|  | RandL | 0.00 ± 0.00 | 0.00 ± 0.00 | 0.00 ± 0.00 | 0.00 ± 0.00 | 0.00 ± 0.00 | 0.00 ± 0.00 |
|  | Bad Teacher | 13.75 ± 10.00 | 10.62 ± 4.24 | 10.62 ± 3.19 | 11.72 ± 2.92 | 10.70 ± 1.09 | 10.78 ± 1.71 |
|  | Amnesiac | 0.00 ± 0.00 | 0.00 ± 0.00 | 0.00 ± 0.00 | 0.00 ± 0.00 | 0.00 ± 0.00 | 0.00 ± 0.00 |
|  | $\delta$-Targeted[1] | 0.00 ± 0.00 | 0.00 ± 0.00 | 0.00 ± 0.00 | 0.00 ± 0.00 | 0.00 ± 0.00 | 0.00 ± 0.00 |
|  | $\delta$-Targeted[3] | 0.00 ± 0.00 | 0.00 ± 0.00 | 0.00 ± 0.00 | 0.00 ± 0.00 | 0.00 ± 0.00 | 0.00 ± 0.00 |
| $D_r$ | *BEFORE* | 100.00 ± 0.00 | 100.00 ± 0.00 | 100.00 ± 0.00 | 100.00 ± 0.00 | 100.00 ± 0.00 | 100.00 ± 0.00 |
|  | NegGrad | 76.01 ± 4.78 | 64.18 ± 3.28 | 38.01 ± 6.06 | 17.06 ± 4.24 | 8.59 ± 1.43 | 4.94 ± 0.98 |
|  | RandL | 92.51 ± 1.67 | 89.36 ± 2.70 | 79.43 ± 3.25 | 64.72 ± 4.14 | 47.95 ± 4.95 | 33.31 ± 2.80 |
|  | Bad Teacher | 70.56 ± 2.01 | 68.54 ± 3.39 | 65.62 ± 1.04 | 54.88 ± 4.27 | 36.45 ± 3.05 | 23.18 ± 4.47 |
|  | Amnesiac | 70.81 ± 0.30 | 70.05 ± 1.37 | 69.05 ± 1.63 | 64.36 ± 2.30 | 43.60 ± 1.78 | 26.51 ± 0.88 |
|  | $\delta$-Targeted[1] | **98.92 ± 0.44** | **98.55 ± 0.51** | 96.54 ± 0.73 | 90.86 ± 1.42 | 80.69 ± 3.27 | 58.80 ± 2.02 |
|  | $\delta$-Targeted[3] | 96.31 ± 0.74 | 96.96 ± 0.95 | **97.23 ± 0.94** | **96.36 ± 0.91** | **92.09 ± 0.78** | **75.74 ± 0.82** |
| $D_t$ | *BEFORE* | 94.02 ± 0.00 | 94.02 ± 0.00 | 94.02 ± 0.00 | 94.02 ± 0.00 | 94.02 ± 0.00 | 94.02 ± 0.00 |
|  | NegGrad | 70.31 ± 4.60 | 59.22 ± 3.17 | 34.67 ± 5.74 | 15.26 ± 3.84 | 8.03 ± 1.40 | 4.53 ± 0.81 |
|  | RandL | 85.60 ± 1.69 | 82.03 ± 2.49 | 72.26 ± 3.25 | 58.06 ± 3.52 | 42.72 ± 4.09 | 29.52 ± 2.69 |
|  | Bad Teacher | 68.93 ± 2.02 | 66.55 ± 3.42 | 64.12 ± 1.23 | 53.19 ± 4.25 | 35.59 ± 3.11 | 22.50 ± 4.55 |
|  | Amnesiac | 69.17 ± 0.40 | 68.22 ± 1.61 | 66.92 ± 1.61 | 62.33 ± 1.88 | 42.08 ± 2.14 | 25.42 ± 1.56 |
|  | $\delta$-Targeted[1] | **91.55 ± 0.39** | **91.15 ± 0.42** | 88.92 ± 0.74 | 83.04 ± 1.44 | 72.93 ± 3.24 | 52.51 ± 2.22 |
|  | $\delta$-Targeted[3] | 89.04 ± 0.61 | 89.75 ± 0.78 | **89.90 ± 0.91** | **88.90 ± 0.98** | **84.37 ± 1.01** | **68.58 ± 0.88** |

**Table 7: Means and standard deviations (percentage) of classification accuracy on CIFAR10 with ViT.**

|  | Method | $|D_f|$=16 | $|D_f|$=32 | $|D_f|$=64 | $|D_f|$=128 | $|D_f|$=256 | $|D_f|$=512 |
|---|---|---|---|---|---|---|---|
| $D_f$ | *BEFORE* | 93.75 ± 6.85 | 94.38 ± 3.64 | 95.00 ± 3.62 | 95.16 ± 2.59 | 94.77 ± 2.27 | 93.48 ± 1.33 |
|  | NegGrad | 8.75 ± 10.90 | 10.00 ± 2.34 | 8.75 ± 2.12 | 9.38 ± 2.21 | 10.86 ± 1.43 | 10.43 ± 1.41 |
|  | RandL | 0.00 ± 0.00 | 0.00 ± 0.00 | 0.00 ± 0.00 | 0.00 ± 0.00 | 0.00 ± 0.00 | 0.00 ± 0.00 |
|  | Bad Teacher | 12.50 ± 5.59 | 6.88 ± 4.59 | 12.19 ± 4.57 | 15.78 ± 5.49 | 18.59 ± 2.86 | 25.94 ± 2.20 |
|  | Amnesiac | 0.00 ± 0.00 | 0.00 ± 0.00 | 0.00 ± 0.00 | 0.00 ± 0.00 | 0.00 ± 0.00 | 0.00 ± 0.00 |
|  | $\delta$-Targeted[1] | 0.00 ± 0.00 | 0.00 ± 0.00 | 0.00 ± 0.00 | 0.00 ± 0.00 | 0.00 ± 0.00 | 0.00 ± 0.00 |
|  | $\delta$-Targeted[3] | 0.00 ± 0.00 | 0.00 ± 0.00 | 0.00 ± 0.00 | 0.00 ± 0.00 | 0.55 ± 0.40 | 2.70 ± 1.01 |
| $D_r$ | *BEFORE* | 93.64 ± 0.00 | 93.64 ± 0.01 | 93.64 ± 0.01 | 93.64 ± 0.01 | 93.63 ± 0.01 | 93.64 ± 0.01 |
|  | NegGrad | 40.34 ± 8.32 | 21.06 ± 6.47 | 14.10 ± 7.21 | 10.05 ± 0.10 | 9.99 ± 0.01 | 9.99 ± 0.01 |
|  | RandL | 79.26 ± 1.48 | 74.43 ± 1.08 | 61.48 ± 3.34 | 47.20 ± 4.53 | 29.39 ± 1.79 | 18.96 ± 2.11 |
|  | Bad Teacher | 85.98 ± 0.44 | 84.69 ± 0.55 | 80.46 ± 1.07 | 75.41 ± 0.97 | 65.98 ± 1.38 | 53.28 ± 3.38 |
|  | Amnesiac | 81.04 ± 0.32 | 78.16 ± 0.53 | 74.15 ± 1.13 | 68.07 ± 1.19 | 58.34 ± 0.83 | 40.50 ± 2.02 |
|  | $\delta$-Targeted[1] | 85.74 ± 3.65 | 83.81 ± 2.10 | 74.86 ± 3.57 | 70.53 ± 3.32 | 65.69 ± 2.58 | 55.62 ± 2.66 |
|  | $\delta$-Targeted[3] | **87.21 ± 1.78** | **85.96 ± 1.10** | **81.13 ± 2.18** | **78.66 ± 1.93** | **75.13 ± 2.29** | **67.42 ± 1.52** |
| $D_t$ | *BEFORE* | 81.69 ± 0.00 | 81.69 ± 0.00 | 81.69 ± 0.00 | 81.69 ± 0.00 | 81.69 ± 0.00 | 81.69 ± 0.00 |
|  | NegGrad | 38.11 ± 7.29 | 20.33 ± 5.93 | 13.80 ± 6.51 | 10.03 ± 0.11 | 10.00 ± 0.12 | 9.91 ± 0.16 |
|  | RandL | 70.53 ± 1.21 | 65.85 ± 0.86 | 54.26 ± 3.02 | 42.25 ± 3.84 | 26.26 ± 1.69 | 16.90 ± 1.63 |
|  | Bad Teacher | **77.51 ± 0.52** | **76.52 ± 0.45** | **72.53 ± 0.82** | 68.53 ± 0.59 | 60.51 ± 1.21 | 50.43 ± 2.82 |
|  | Amnesiac | 72.48 ± 0.30 | 69.71 ± 0.48 | 66.16 ± 1.09 | 60.93 ± 0.87 | 52.44 ± 0.83 | 36.84 ± 1.96 |
|  | $\delta$-Targeted[1] | 75.47 ± 2.67 | 73.91 ± 1.47 | 66.41 ± 2.81 | 62.63 ± 2.84 | 58.08 ± 2.25 | 49.50 ± 2.40 |
|  | $\delta$-Targeted[3] | 76.68 ± 1.25 | 75.84 ± 0.81 | 71.66 ± 1.76 | **69.43 ± 1.44** | **66.20 ± 1.97** | **59.58 ± 1.25** |

## B.2 Performance on CIFAR100 with Different Neural Network Structures

**Table 8: Means and standard deviations (percentage) of classification accuracy on CIFAR100 with ResNet-18.**

|  | Method | $|D_f|$=16 | $|D_f|$=32 | $|D_f|$=64 | $|D_f|$=128 | $|D_f|$=256 | $|D_f|$=512 |
|---|---|---|---|---|---|---|---|
| $D_f$ | *BEFORE* | 100.00 ± 0.00 | 100.00 ± 0.00 | 100.00 ± 0.00 | 99.84 ± 0.31 | 99.92 ± 0.16 | 99.96 ± 0.08 |
| | NegGrad | 0.00 ± 0.00 | 0.00 ± 0.00 | 0.00 ± 0.00 | 20.00 ± 40.00 | 0.00 ± 0.00 | 0.00 ± 0.00 |
| | RandL | 0.00 ± 0.00 | 0.00 ± 0.00 | 0.00 ± 0.00 | 0.00 ± 0.00 | 0.00 ± 0.00 | 0.00 ± 0.00 |
| | Bad Teacher | 0.00 ± 0.00 | 0.62 ± 1.25 | 0.94 ± 1.25 | 0.47 ± 0.62 | 1.33 ± 0.19 | 0.86 ± 0.23 |
| | Amnesiac | 0.00 ± 0.00 | 0.00 ± 0.00 | 0.00 ± 0.00 | 0.00 ± 0.00 | 0.00 ± 0.00 | 0.00 ± 0.00 |
| | $\delta$-Targeted[1] | 0.00 ± 0.00 | 0.00 ± 0.00 | 0.00 ± 0.00 | 0.00 ± 0.00 | 0.00 ± 0.00 | 0.00 ± 0.00 |
| | $\delta$-Targeted[3] | 0.00 ± 0.00 | 0.00 ± 0.00 | 0.00 ± 0.00 | 0.00 ± 0.00 | 0.00 ± 0.00 | 0.00 ± 0.00 |
| $D_r$ | *BEFORE* | 99.96 ± 0.00 | 99.96 ± 0.00 | 99.96 ± 0.00 | 99.96 ± 0.00 | 99.96 ± 0.00 | 99.96 ± 0.00 |
| | NegGrad | 83.71 ± 3.40 | 86.98 ± 0.84 | 84.97 ± 1.28 | 83.83 ± 8.28 | 70.57 ± 1.72 | 55.92 ± 0.95 |
| | RandL | 91.86 ± 2.90 | 95.32 ± 0.86 | 95.40 ± 0.57 | 94.58 ± 1.13 | 92.86 ± 0.40 | 89.79 ± 0.54 |
| | Bad Teacher | 36.13 ± 1.50 | 36.75 ± 1.51 | 35.45 ± 1.66 | 29.34 ± 2.15 | 25.57 ± 2.45 | 16.33 ± 2.24 |
| | Amnesiac | 35.47 ± 1.62 | 34.69 ± 0.65 | 33.22 ± 2.57 | 31.17 ± 0.89 | 22.39 ± 0.88 | 11.80 ± 0.63 |
| | $\delta$-Targeted[1] | **98.17 ± 0.90** | **98.99 ± 0.35** | **98.84 ± 0.10** | **98.16 ± 0.71** | **97.34 ± 0.37** | 94.53 ± 0.56 |
| | $\delta$-Targeted[3] | 94.89 ± 2.10 | 97.10 ± 0.69 | 97.05 ± 0.46 | 96.49 ± 1.05 | 96.06 ± 0.20 | **95.36 ± 0.41** |
| $D_t$ | *BEFORE* | 71.94 ± 0.00 | 71.94 ± 0.00 | 71.94 ± 0.00 | 71.94 ± 0.00 | 71.94 ± 0.00 | 71.94 ± 0.00 |
| | NegGrad | 56.30 ± 2.06 | 57.75 ± 0.63 | 56.30 ± 1.00 | 56.35 ± 7.57 | 46.14 ± 1.10 | 36.38 ± 0.44 |
| | RandL | 60.95 ± 2.05 | 64.02 ± 0.57 | 64.28 ± 0.36 | 63.21 ± 0.71 | 60.74 ± 0.46 | 56.39 ± 0.23 |
| | Bad Teacher | 30.66 ± 0.90 | 31.33 ± 1.25 | 30.04 ± 1.35 | 24.86 ± 1.82 | 21.32 ± 1.97 | 13.97 ± 1.77 |
| | Amnesiac | 30.54 ± 1.54 | 29.85 ± 0.98 | 28.35 ± 2.16 | 26.93 ± 0.94 | 19.26 ± 0.96 | 10.08 ± 0.42 |
| | $\delta$-Targeted[1] | **66.27 ± 1.38** | **68.06 ± 0.52** | **68.08 ± 0.16** | **67.36 ± 0.38** | **66.08 ± 0.30** | 62.87 ± 0.71 |
| | $\delta$-Targeted[3] | 63.38 ± 1.72 | 65.89 ± 0.60 | 66.52 ± 0.35 | 66.22 ± 0.60 | 65.45 ± 0.23 | **64.06 ± 0.55** |

**Table 9: Means and standard deviations (percentage) of classification accuracy on CIFAR100 with ResNet-50.**

|  | Method | $|D_f|$=16 | $|D_f|$=32 | $|D_f|$=64 | $|D_f|$=128 | $|D_f|$=256 | $|D_f|$=512 |
|---|---|---|---|---|---|---|---|
| $D_f$ | *BEFORE* | 100.00 ± 0.00 | 100.00 ± 0.00 | 100.00 ± 0.00 | 99.84 ± 0.31 | 99.92 ± 0.16 | 99.96 ± 0.08 |
| | NegGrad | 0.00 ± 0.00 | 0.00 ± 0.00 | 0.00 ± 0.00 | 0.00 ± 0.00 | 0.00 ± 0.00 | 0.00 ± 0.00 |
| | RandL | 0.00 ± 0.00 | 0.00 ± 0.00 | 0.00 ± 0.00 | 0.00 ± 0.00 | 0.00 ± 0.00 | 0.00 ± 0.00 |
| | Bad Teacher | 0.00 ± 0.00 | 0.62 ± 1.25 | 1.25 ± 1.17 | 1.25 ± 0.80 | 1.33 ± 0.19 | 1.41 ± 0.19 |
| | Amnesiac | 0.00 ± 0.00 | 0.00 ± 0.00 | 0.00 ± 0.00 | 0.00 ± 0.00 | 0.00 ± 0.00 | 0.00 ± 0.00 |
| | $\delta$-Targeted[1] | 0.00 ± 0.00 | 0.00 ± 0.00 | 0.00 ± 0.00 | 0.00 ± 0.00 | 0.00 ± 0.00 | 0.00 ± 0.00 |
| | $\delta$-Targeted[3] | 0.00 ± 0.00 | 0.00 ± 0.00 | 0.00 ± 0.00 | 0.00 ± 0.00 | 0.00 ± 0.00 | 0.00 ± 0.00 |
| $D_r$ | *BEFORE* | 99.97 ± 0.00 | 99.97 ± 0.00 | 99.97 ± 0.00 | 99.97 ± 0.00 | 99.97 ± 0.00 | 99.97 ± 0.00 |
| | NegGrad | 66.36 ± 7.53 | 66.55 ± 5.26 | 60.86 ± 3.08 | 57.38 ± 3.42 | 51.95 ± 3.63 | 43.16 ± 1.36 |
| | RandL | 87.54 ± 6.39 | 91.20 ± 2.52 | 92.83 ± 1.95 | 93.29 ± 1.47 | 92.44 ± 0.53 | 89.08 ± 0.85 |
| | Bad Teacher | 23.47 ± 1.07 | 23.83 ± 2.11 | 22.25 ± 0.71 | 11.25 ± 1.07 | 5.22 ± 0.82 | 3.36 ± 0.51 |
| | Amnesiac | 24.92 ± 0.98 | 25.12 ± 1.31 | 24.37 ± 0.62 | 21.51 ± 1.67 | 12.26 ± 1.60 | 7.36 ± 0.56 |
| | $\delta$-Targeted[1] | **98.36 ± 1.39** | **97.07 ± 1.26** | **97.27 ± 1.06** | **97.36 ± 0.66** | **96.87 ± 0.63** | 93.58 ± 0.74 |
| | $\delta$-Targeted[3] | 93.71 ± 3.49 | 94.68 ± 1.86 | 95.37 ± 1.43 | 95.55 ± 1.36 | 95.46 ± 0.53 | **94.27 ± 0.67** |
| $D_t$ | *BEFORE* | 74.67 ± 0.00 | 74.67 ± 0.00 | 74.67 ± 0.00 | 74.67 ± 0.00 | 74.67 ± 0.00 | 74.67 ± 0.00 |
| | NegGrad | 45.94 ± 4.63 | 45.44 ± 3.18 | 41.33 ± 2.02 | 38.76 ± 2.07 | 35.34 ± 2.76 | 29.79 ± 1.02 |
| | RandL | 60.14 ± 4.44 | 62.62 ± 2.06 | 64.22 ± 1.76 | 64.39 ± 1.40 | 62.79 ± 0.48 | 58.55 ± 0.41 |
| | Bad Teacher | 21.45 ± 0.90 | 21.28 ± 2.17 | 20.22 ± 0.75 | 9.70 ± 1.08 | 4.41 ± 0.74 | 2.62 ± 0.56 |
| | Amnesiac | 22.30 ± 0.69 | 22.33 ± 1.07 | 21.98 ± 0.69 | 19.08 ± 1.49 | 10.91 ± 1.58 | 6.56 ± 0.77 |
| | $\delta$-Targeted[1] | **69.77 ± 2.57** | **67.91 ± 1.18** | **68.89 ± 1.01** | **69.18 ± 0.87** | **68.55 ± 0.67** | 64.65 ± 0.49 |
| | $\delta$-Targeted[3] | 64.80 ± 3.68 | 65.76 ± 1.73 | 67.09 ± 1.40 | 67.48 ± 1.47 | 67.26 ± 0.43 | **65.51 ± 0.60** |

**Table 10: Means and standard deviations (percentage) of classification accuracy on CIFAR100 with ViT.**

| | Method | $|D_f|$=16 | $|D_f|$=32 | $|D_f|$=64 | $|D_f|$=128 | $|D_f|$=256 | $|D_f|$=512 |
|---|---|---|---|---|---|---|---|
| $D_f$ | *BEFORE* | 86.25 ± 8.29 | 81.88 ± 8.93 | 83.44 ± 2.54 | 84.84 ± 2.19 | 84.30 ± 0.62 | 84.84 ± 1.35 |
| | NegGrad | 1.25 ± 2.50 | 0.62 ± 1.25 | 0.62 ± 0.77 | 1.09 ± 0.38 | 0.78 ± 0.49 | 0.70 ± 0.47 |
| | RandL | 0.00 ± 0.00 | 0.00 ± 0.00 | 0.00 ± 0.00 | 0.00 ± 0.00 | 0.00 ± 0.00 | 0.00 ± 0.00 |
| | Bad Teacher | 3.75 ± 5.00 | 2.50 ± 2.34 | 4.06 ± 1.88 | 5.00 ± 1.45 | 6.72 ± 0.76 | 10.35 ± 0.54 |
| | Amnesiac | 0.00 ± 0.00 | 0.00 ± 0.00 | 0.00 ± 0.00 | 0.00 ± 0.00 | 0.00 ± 0.00 | 0.00 ± 0.00 |
| | $\delta$-Targeted[1] | 0.00 ± 0.00 | 0.00 ± 0.00 | 0.00 ± 0.00 | 0.00 ± 0.00 | 0.00 ± 0.00 | 0.00 ± 0.00 |
| | $\delta$-Targeted[3] | 0.00 ± 0.00 | 0.00 ± 0.00 | 0.00 ± 0.00 | 0.00 ± 0.00 | 0.00 ± 0.00 | 0.00 ± 0.00 |
| $D_r$ | *BEFORE* | 85.18 ± 0.01 | 85.18 ± 0.01 | 85.19 ± 0.02 | 85.18 ± 0.00 | 85.19 ± 0.00 | 85.19 ± 0.01 |
| | NegGrad | 38.64 ± 2.76 | 25.91 ± 1.68 | 13.58 ± 3.10 | 1.55 ± 1.09 | 1.00 ± 0.00 | 1.00 ± 0.01 |
| | RandL | 66.80 ± 3.46 | 63.40 ± 2.19 | 56.68 ± 1.36 | 47.10 ± 2.25 | 36.13 ± 2.18 | 22.14 ± 0.82 |
| | Bad Teacher | 74.08 ± 0.38 | 71.23 ± 0.60 | 63.16 ± 1.72 | 52.82 ± 0.80 | 39.30 ± 1.25 | 26.99 ± 1.19 |
| | Amnesiac | 64.19 ± 1.46 | 62.44 ± 1.16 | 57.94 ± 1.15 | 53.96 ± 0.99 | 46.12 ± 0.75 | 31.17 ± 1.18 |
| | $\delta$-Targeted[1] | 83.65 ± 0.52 | 82.49 ± 1.10 | 79.90 ± 0.99 | 76.41 ± 1.00 | 73.74 ± 1.49 | 70.48 ± 1.16 |
| | $\delta$-Targeted[3] | **84.04 ± 0.74** | **82.97 ± 1.15** | **80.86 ± 0.71** | **78.79 ± 0.71** | **77.57 ± 1.36** | **75.54 ± 0.65** |
| $D_t$ | *BEFORE* | 54.59 ± 0.00 | 54.59 ± 0.00 | 54.59 ± 0.00 | 54.59 ± 0.00 | 54.59 ± 0.00 | 54.59 ± 0.00 |
| | NegGrad | 28.75 ± 1.70 | 20.25 ± 1.36 | 11.22 ± 2.61 | 1.45 ± 0.86 | 1.01 ± 0.06 | 0.98 ± 0.00 |
| | RandL | 45.19 ± 1.59 | 42.82 ± 1.23 | 38.65 ± 1.03 | 32.59 ± 1.49 | 25.17 ± 1.46 | 15.58 ± 0.43 |
| | Bad Teacher | 50.66 ± 0.48 | 49.06 ± 0.30 | 44.50 ± 0.66 | 38.84 ± 0.74 | 30.35 ± 1.05 | 22.14 ± 0.85 |
| | Amnesiac | 43.38 ± 0.59 | 42.34 ± 0.61 | 39.70 ± 0.59 | 37.25 ± 0.78 | 31.76 ± 0.50 | 22.45 ± 0.57 |
| | $\delta$-Targeted[1] | 53.69 ± 0.20 | 53.25 ± 0.49 | 51.63 ± 0.59 | 49.86 ± 0.41 | 47.92 ± 0.94 | 46.00 ± 0.78 |
| | $\delta$-Targeted[3] | **53.92 ± 0.24** | **53.37 ± 0.36** | **52.14 ± 0.48** | **51.31 ± 0.32** | **50.21 ± 0.70** | **48.73 ± 0.70** |

## B.3 Performance on Fashion-MNIST with Different Neural Network Structures

**Table 11: Means and standard deviations (percentage) of classification accuracy on Fashion-MNIST with ResNet-18.**

| | Method | $|D_f|$=16 | $|D_f|$=32 | $|D_f|$=64 | $|D_f|$=128 | $|D_f|$=256 | $|D_f|$=512 |
|---|---|---|---|---|---|---|---|
| $D_f$ | *BEFORE* | 95.00 ± 4.68 | 94.38 ± 3.64 | 94.69 ± 1.59 | 95.47 ± 0.91 | 96.02 ± 0.67 | 96.48 ± 0.86 |
| | NegGrad | 0.00 ± 0.00 | 0.00 ± 0.00 | 0.00 ± 0.00 | 0.00 ± 0.00 | 0.00 ± 0.00 | 0.00 ± 0.00 |
| | RandL | 0.00 ± 0.00 | 0.00 ± 0.00 | 0.00 ± 0.00 | 0.00 ± 0.00 | 0.00 ± 0.00 | 0.00 ± 0.00 |
| | Bad Teacher | 12.50 ± 6.85 | 6.88 ± 2.34 | 8.44 ± 4.38 | 9.84 ± 3.51 | 12.89 ± 3.62 | 13.95 ± 5.09 |
| | Amnesiac | 0.00 ± 0.00 | 0.00 ± 0.00 | 0.00 ± 0.00 | 0.00 ± 0.00 | 0.94 ± 1.37 | 3.24 ± 1.88 |
| | $\delta$-Targeted[1] | 0.00 ± 0.00 | 0.00 ± 0.00 | 0.00 ± 0.00 | 0.00 ± 0.00 | 0.00 ± 0.00 | 0.00 ± 0.00 |
| | $\delta$-Targeted[3] | 0.00 ± 0.00 | 0.00 ± 0.00 | 0.00 ± 0.00 | 0.00 ± 0.00 | 0.00 ± 0.00 | 0.00 ± 0.00 |
| $D_r$ | *BEFORE* | 96.90 ± 0.00 | 96.90 ± 0.00 | 96.90 ± 0.01 | 96.90 ± 0.00 | 96.90 ± 0.00 | 96.90 ± 0.01 |
| | NegGrad | 73.65 ± 1.36 | 61.92 ± 3.62 | 45.79 ± 3.52 | 26.79 ± 0.93 | 16.05 ± 1.11 | 8.77 ± 1.49 |
| | RandL | 86.51 ± 1.35 | 85.38 ± 1.46 | 75.68 ± 1.65 | 60.56 ± 1.10 | 43.32 ± 1.95 | 28.76 ± 1.22 |
| | Bad Teacher | 86.49 ± 0.40 | 86.07 ± 0.29 | 85.71 ± 0.52 | 80.54 ± 2.86 | 68.33 ± 2.65 | 55.75 ± 3.67 |
| | Amnesiac | 85.68 ± 0.40 | 84.32 ± 0.77 | 81.41 ± 1.03 | 76.38 ± 1.07 | 59.85 ± 3.75 | 42.45 ± 3.57 |
| | $\delta$-Targeted[1] | **93.94 ± 0.50** | 92.13 ± 0.68 | 87.06 ± 0.27 | 76.28 ± 1.39 | 59.01 ± 3.18 | 39.59 ± 1.16 |
| | $\delta$-Targeted[3] | 92.93 ± 0.98 | **93.35 ± 0.41** | **92.40 ± 0.57** | **87.61 ± 1.21** | **76.71 ± 1.94** | **58.28 ± 2.43** |
| $D_t$ | *BEFORE* | 91.65 ± 0.00 | 91.65 ± 0.00 | 91.65 ± 0.00 | 91.65 ± 0.00 | 91.65 ± 0.00 | 91.65 ± 0.00 |
| | NegGrad | 69.50 ± 1.73 | 58.26 ± 4.21 | 42.21 ± 3.13 | 23.90 ± 0.98 | 13.82 ± 1.02 | 7.57 ± 1.16 |
| | RandL | 82.03 ± 1.07 | 81.17 ± 1.37 | 71.07 ± 2.18 | 56.25 ± 1.06 | 39.02 ± 2.11 | 24.99 ± 1.30 |
| | Bad Teacher | 85.18 ± 0.34 | 84.56 ± 0.36 | 84.27 ± 0.40 | 78.98 ± 2.81 | 67.08 ± 2.44 | **55.04 ± 3.51** |
| | Amnesiac | 84.37 ± 0.56 | 83.07 ± 0.54 | 80.05 ± 1.17 | 74.67 ± 0.90 | 58.42 ± 3.83 | 41.74 ± 3.85 |
| | $\delta$-Targeted[1] | **88.91 ± 0.48** | 87.35 ± 0.70 | 82.36 ± 0.48 | 71.60 ± 2.03 | 54.48 ± 3.36 | 35.16 ± 1.24 |
| | $\delta$-Targeted[3] | 88.12 ± 0.79 | **88.57 ± 0.37** | **87.42 ± 0.58** | **82.74 ± 1.09** | **72.01 ± 1.92** | 53.66 ± 2.49 |

 

**Table 12: Means and standard deviations (percentage) of classification accuracy on Fashion-MNIST with ResNet-50.**

| | Method | $|D_f|$=16 | $|D_f|$=32 | $|D_f|$=64 | $|D_f|$=128 | $|D_f|$=256 | $|D_f|$=512 |
|---|---|---|---|---|---|---|---|
| $D_f$ | *BEFORE* | 100.00 ± 0.00 | 98.75 ± 1.53 | 97.81 ± 1.59 | 97.66 ± 1.48 | 96.95 ± 1.06 | 97.30 ± 0.52 |
| | NegGrad | 0.00 ± 0.00 | 0.00 ± 0.00 | 0.00 ± 0.00 | 0.00 ± 0.00 | 0.00 ± 0.00 | 0.00 ± 0.00 |
| | RandL | 0.00 ± 0.00 | 0.00 ± 0.00 | 0.00 ± 0.00 | 0.00 ± 0.00 | 0.00 ± 0.00 | 0.00 ± 0.00 |
| | Bad Teacher | 11.25 ± 7.29 | 11.25 ± 4.24 | 11.88 ± 2.90 | 10.94 ± 2.47 | 13.59 ± 1.79 | 12.66 ± 2.05 |
| | Amnesiac | 0.00 ± 0.00 | 0.00 ± 0.00 | 0.94 ± 1.88 | 0.47 ± 0.94 | 6.25 ± 4.98 | 3.05 ± 1.36 |
| | $\delta$-Targeted[1] | 0.00 ± 0.00 | 0.00 ± 0.00 | 0.00 ± 0.00 | 0.00 ± 0.00 | 0.00 ± 0.00 | 0.00 ± 0.00 |
| | $\delta$-Targeted[3] | 0.00 ± 0.00 | 0.00 ± 0.00 | 0.00 ± 0.00 | 0.00 ± 0.00 | 0.00 ± 0.00 | 0.00 ± 0.00 |
| $D_r$ | *BEFORE* | 97.10 ± 0.00 | 97.10 ± 0.00 | 97.10 ± 0.00 | 97.10 ± 0.00 | 97.10 ± 0.00 | 97.10 ± 0.00 |
| | NegGrad | 68.70 ± 5.09 | 51.92 ± 6.19 | 34.08 ± 3.29 | 18.40 ± 1.46 | 9.01 ± 0.71 | 5.29 ± 0.41 |
| | RandL | 81.25 ± 6.89 | 75.48 ± 3.47 | 64.52 ± 4.04 | 46.83 ± 2.00 | 29.98 ± 1.67 | 20.67 ± 0.90 |
| | Bad Teacher | 83.84 ± 0.50 | 83.03 ± 1.65 | 77.70 ± 2.68 | 69.85 ± 4.52 | 51.28 ± 5.79 | 36.32 ± 3.92 |
| | Amnesiac | 83.74 ± 1.30 | 82.68 ± 0.98 | 78.56 ± 1.80 | 73.40 ± 0.83 | 60.89 ± 2.75 | 39.38 ± 1.59 |
| | $\delta$-Targeted[1] | **94.11 ± 0.63** | 88.65 ± 2.09 | 80.80 ± 2.20 | 68.19 ± 3.23 | 51.23 ± 3.56 | 34.98 ± 1.17 |
| | $\delta$-Targeted[3] | 93.96 ± 1.11 | **93.56 ± 0.34** | **90.87 ± 1.12** | **83.50 ± 2.48** | **69.57 ± 1.73** | **49.50 ± 0.85** |
| $D_t$ | *BEFORE* | 91.79 ± 0.00 | 91.79 ± 0.00 | 91.79 ± 0.00 | 91.79 ± 0.00 | 91.79 ± 0.00 | 91.79 ± 0.00 |
| | NegGrad | 64.50 ± 5.13 | 47.82 ± 6.14 | 30.65 ± 3.27 | 15.94 ± 1.63 | 7.76 ± 0.61 | 4.53 ± 0.26 |
| | RandL | 76.48 ± 6.88 | 70.32 ± 3.42 | 59.76 ± 4.00 | 42.36 ± 1.81 | 26.24 ± 1.62 | 17.54 ± 1.10 |
| | Bad Teacher | 82.79 ± 0.52 | 81.58 ± 1.94 | 76.59 ± 2.54 | 69.00 ± 4.55 | 50.54 ± 5.64 | 35.94 ± 3.93 |
| | Amnesiac | 82.81 ± 1.29 | 81.62 ± 1.12 | 77.85 ± 1.79 | 72.95 ± 0.79 | 60.62 ± 2.94 | 38.86 ± 1.98 |
| | $\delta$-Targeted[1] | 89.19 ± 0.53 | 83.47 ± 2.55 | 75.66 ± 2.41 | 62.94 ± 3.07 | 46.41 ± 3.53 | 31.12 ± 0.94 |
| | $\delta$-Targeted[3] | **89.44 ± 0.73** | **88.87 ± 0.35** | **86.05 ± 1.28** | **78.45 ± 2.57** | **64.35 ± 1.61** | **45.15 ± 0.77** |

**Table 13: Means and standard deviations (percentage) of classification accuracy on Fashion-MNIST with ViT.**

| | Method | $|D_f|$=16 | $|D_f|$=32 | $|D_f|$=64 | $|D_f|$=128 | $|D_f|$=256 | $|D_f|$=512 |
|---|---|---|---|---|---|---|---|
| $D_f$ | *BEFORE* | 93.75 ± 5.59 | 93.12 ± 4.59 | 91.88 ± 2.69 | 91.72 ± 1.17 | 92.42 ± 0.58 | 93.48 ± 0.64 |
| | NegGrad | 8.75 ± 9.35 | 8.12 ± 9.19 | 10.94 ± 6.63 | 6.72 ± 5.13 | 10.08 ± 2.87 | 10.43 ± 1.51 |
| | RandL | 0.00 ± 0.00 | 0.00 ± 0.00 | 0.00 ± 0.00 | 0.00 ± 0.00 | 0.00 ± 0.00 | 1.37 ± 0.35 |
| | Bad Teacher | 0.00 ± 0.00 | 0.00 ± 0.00 | 0.00 ± 0.00 | 0.00 ± 0.00 | 0.00 ± 0.00 | 0.00 ± 0.00 |
| | Amnesiac | 0.00 ± 0.00 | 1.25 ± 2.50 | 6.56 ± 2.50 | 18.44 ± 3.37 | 29.92 ± 2.82 | 25.74 ± 2.58 |
| | $\delta$-Targeted[1] | 0.00 ± 0.00 | 0.00 ± 0.00 | 1.88 ± 1.17 | 3.28 ± 1.94 | 13.67 ± 3.91 | 52.50 ± 4.18 |
| | $\delta$-Targeted[3] | 33.75 ± 8.48 | 31.25 ± 6.56 | 46.88 ± 6.01 | 67.19 ± 0.49 | 76.64 ± 2.35 | 77.54 ± 2.08 |
| $D_r$ | *BEFORE* | 94.11 ± 0.00 | 94.11 ± 0.00 | 94.11 ± 0.01 | 94.11 ± 0.00 | 94.11 ± 0.00 | 94.11 ± 0.01 |
| | NegGrad | 19.60 ± 7.44 | 9.94 ± 1.37 | 11.50 ± 4.18 | 8.39 ± 3.88 | 10.00 ± 0.01 | 10.00 ± 0.01 |
| | RandL | 48.39 ± 6.41 | 31.38 ± 5.46 | 20.56 ± 4.46 | 13.27 ± 2.49 | 8.74 ± 2.35 | 7.19 ± 1.05 |
| | Bad Teacher | 67.16 ± 9.11 | 59.51 ± 4.55 | 50.04 ± 3.26 | 39.13 ± 2.33 | 27.45 ± 1.21 | 21.05 ± 0.66 |
| | Amnesiac | 75.10 ± 6.75 | 73.45 ± 3.09 | 64.50 ± 2.36 | 57.24 ± 1.96 | 48.30 ± 1.54 | 38.16 ± 1.86 |
| | $\delta$-Targeted[1] | 87.21 ± 0.54 | 84.90 ± 1.33 | 80.34 ± 1.15 | 73.08 ± 1.23 | 62.07 ± 1.20 | 61.94 ± 2.66 |
| | $\delta$-Targeted[3] | **90.53 ± 0.31** | **89.02 ± 0.67** | **86.85 ± 0.67** | **84.00 ± 0.38** | **80.82 ± 1.02** | **77.31 ± 1.44** |
| $D_t$ | *BEFORE* | 92.41 ± 0.00 | 92.41 ± 0.00 | 92.41 ± 0.00 | 92.41 ± 0.00 | 92.41 ± 0.00 | 92.41 ± 0.00 |
| | NegGrad | 19.58 ± 7.47 | 10.03 ± 1.46 | 11.58 ± 4.10 | 8.51 ± 3.93 | 10.15 ± 0.12 | 10.06 ± 0.19 |
| | RandL | 47.47 ± 6.61 | 30.98 ± 5.32 | 20.20 ± 4.20 | 13.16 ± 2.61 | 8.71 ± 2.08 | 6.87 ± 0.96 |
| | Bad Teacher | 65.85 ± 9.02 | 58.35 ± 4.44 | 48.81 ± 3.24 | 37.86 ± 2.46 | 26.34 ± 1.54 | 20.44 ± 0.78 |
| | Amnesiac | 73.54 ± 6.76 | 71.91 ± 3.05 | 62.96 ± 2.56 | 55.70 ± 2.01 | 46.49 ± 1.51 | 36.23 ± 1.94 |
| | $\delta$-Targeted[1] | 85.87 ± 0.77 | 83.72 ± 1.38 | 79.11 ± 1.22 | 71.81 ± 1.36 | 61.22 ± 1.35 | 61.48 ± 2.44 |
| | $\delta$-Targeted[3] | **89.09 ± 0.31** | **87.58 ± 0.71** | **85.36 ± 0.81** | **82.90 ± 0.57** | **80.05 ± 1.07** | **76.80 ± 1.38** |

## B.4  Performance on Tiny-ImageNet with Different Neural Network Structures

Table 14: Means and standard deviations (percentage) of classification accuracy on Tiny-ImageNet with ResNet-18.

|  | Method | $|D_f|$=16 | $|D_f|$=32 | $|D_f|$=64 | $|D_f|$=128 | $|D_f|$=256 | $|D_f|$=512 |
|---|---|---|---|---|---|---|---|
| $D_f$ | *BEFORE* | 85.00 ± 6.37 | 85.62 ± 3.75 | 84.69 ± 2.07 | 84.84 ± 1.06 | 84.06 ± 1.17 | 84.61 ± 0.67 |
|  | NegGrad | 0.00 ± 0.00 | 0.00 ± 0.00 | 0.00 ± 0.00 | 0.00 ± 0.00 | 0.00 ± 0.00 | 0.00 ± 0.00 |
|  | RandL | 0.00 ± 0.00 | 0.00 ± 0.00 | 0.00 ± 0.00 | 0.00 ± 0.00 | 0.00 ± 0.00 | 0.00 ± 0.00 |
|  | Bad Teacher | 1.25 ± 2.50 | 0.62 ± 1.25 | 0.00 ± 0.00 | 0.62 ± 0.58 | 0.31 ± 0.46 | 0.55 ± 0.62 |
|  | Amnesiac | 0.00 ± 0.00 | 0.00 ± 0.00 | 0.00 ± 0.00 | 0.00 ± 0.00 | 0.00 ± 0.00 | 0.00 ± 0.00 |
|  | $\delta$-Targeted[1] | 0.00 ± 0.00 | 0.00 ± 0.00 | 0.00 ± 0.00 | 0.00 ± 0.00 | 0.00 ± 0.00 | 0.00 ± 0.00 |
|  | $\delta$-Targeted[3] | 0.00 ± 0.00 | 0.00 ± 0.00 | 0.00 ± 0.00 | 0.00 ± 0.00 | 0.00 ± 0.00 | 0.04 ± 0.08 |
| $D_r$ | *BEFORE* | 85.01 ± 0.01 | 85.01 ± 0.00 | 85.01 ± 0.00 | 85.01 ± 0.00 | 85.02 ± 0.00 | 85.02 ± 0.00 |
|  | NegGrad | 58.03 ± 1.68 | 61.53 ± 1.86 | 63.25 ± 0.68 | 58.79 ± 0.71 | 53.26 ± 0.56 | 45.46 ± 1.72 |
|  | RandL | 62.68 ± 1.90 | 69.22 ± 2.68 | 74.88 ± 1.21 | 76.67 ± 0.44 | 76.29 ± 0.48 | 73.07 ± 0.48 |
|  | Bad Teacher | 18.82 ± 0.57 | 19.46 ± 0.65 | 19.73 ± 1.07 | 18.01 ± 0.79 | 12.37 ± 1.02 | 8.49 ± 0.56 |
|  | Amnesiac | 12.62 ± 0.64 | 12.67 ± 0.54 | 12.68 ± 0.78 | 12.18 ± 0.57 | 11.63 ± 0.54 | 8.04 ± 0.48 |
|  | $\delta$-Targeted[1] | **83.02 ± 1.07** | **83.43 ± 0.52** | **83.60 ± 0.38** | **83.36 ± 0.26** | **82.93 ± 0.18** | 81.70 ± 0.35 |
|  | $\delta$-Targeted[3] | 76.88 ± 4.04 | 77.96 ± 2.31 | 80.45 ± 0.71 | 81.89 ± 0.33 | 82.77 ± 0.22 | **82.51 ± 0.20** |
| $D_t$ | *BEFORE* | 52.71 ± 0.00 | 52.71 ± 0.00 | 52.71 ± 0.00 | 52.71 ± 0.00 | 52.71 ± 0.00 | 52.71 ± 0.00 |
|  | NegGrad | 40.47 ± 1.05 | 42.26 ± 0.87 | 42.37 ± 0.81 | 39.40 ± 0.77 | 35.90 ± 0.42 | 31.02 ± 0.97 |
|  | RandL | 43.58 ± 1.17 | 46.22 ± 1.31 | 48.25 ± 0.42 | 48.39 ± 0.40 | 47.61 ± 0.49 | 45.15 ± 0.38 |
|  | Bad Teacher | 17.81 ± 0.62 | 18.90 ± 0.51 | 18.87 ± 1.42 | 17.08 ± 1.00 | 11.72 ± 0.97 | 7.88 ± 0.61 |
|  | Amnesiac | 12.22 ± 0.46 | 12.15 ± 0.63 | 12.07 ± 0.72 | 11.40 ± 0.42 | 10.65 ± 0.82 | 7.10 ± 0.62 |
|  | $\delta$-Targeted[1] | **52.00 ± 0.34** | **52.18 ± 0.23** | **52.04 ± 0.24** | **51.75 ± 0.30** | **51.57 ± 0.25** | 50.97 ± 0.35 |
|  | $\delta$-Targeted[3] | 49.63 ± 1.32 | 50.08 ± 0.64 | 50.97 ± 0.49 | 51.24 ± 0.41 | 51.40 ± 0.40 | **51.37 ± 0.20** |

Table 15: Means and standard deviations (percentage) of classification accuracy on Tiny-ImageNet with ResNet-50.

|  | Method | $|D_f|$=16 | $|D_f|$=32 | $|D_f|$=64 | $|D_f|$=128 | $|D_f|$=256 | $|D_f|$=512 |
|---|---|---|---|---|---|---|---|
| $D_f$ | *BEFORE* | 28.12 ± 9.38 | 29.69 ± 4.69 | 28.91 ± 0.78 | 30.86 ± 5.08 | 32.23 ± 3.71 | 33.59 ± 0.00 |
|  | NegGrad | 0.00 ± 0.00 | 0.00 ± 0.00 | 0.00 ± 0.00 | 0.00 ± 0.00 | 0.00 ± 0.00 | 0.00 ± 0.00 |
|  | RandL | 0.00 ± 0.00 | 0.00 ± 0.00 | 0.00 ± 0.00 | 0.00 ± 0.00 | 0.00 ± 0.00 | 0.00 ± 0.00 |
|  | Bad Teacher | 0.00 ± 0.00 | 0.00 ± 0.00 | 0.00 ± 0.00 | 0.31 ± 0.38 | 0.31 ± 0.29 | 0.35 ± 0.19 |
|  | Amnesiac | 0.00 ± 0.00 | 0.00 ± 0.00 | 0.00 ± 0.00 | 0.00 ± 0.00 | 0.00 ± 0.00 | 0.00 ± 0.00 |
|  | $\delta$-Targeted[1] | 0.00 ± 0.00 | 0.00 ± 0.00 | 0.00 ± 0.00 | 0.00 ± 0.00 | 0.00 ± 0.00 | 0.00 ± 0.00 |
|  | $\delta$-Targeted[3] | 0.00 ± 0.00 | 0.00 ± 0.00 | 0.00 ± 0.00 | 0.00 ± 0.00 | 0.00 ± 0.00 | 0.00 ± 0.00 |
| $D_r$ | *BEFORE* | 53.04 ± 0.02 | 53.05 ± 0.00 | 53.05 ± 0.00 | 53.06 ± 0.00 | 53.05 ± 0.00 | 53.06 ± 0.00 |
|  | NegGrad | 38.70 ± 1.30 | 42.01 ± 0.85 | 40.98 ± 0.73 | 36.26 ± 1.20 | 30.46 ± 0.75 | 27.95 ± 0.96 |
|  | RandL | 39.28 ± 0.61 | 38.70 ± 0.88 | 39.13 ± 0.67 | 40.85 ± 0.63 | 42.73 ± 0.38 | 43.96 ± 0.85 |
|  | Bad Teacher | 28.46 ± 0.77 | 27.44 ± 2.26 | 28.75 ± 0.56 | 26.27 ± 0.32 | 21.78 ± 0.28 | 21.22 ± 0.29 |
|  | Amnesiac | 29.35 ± 0.61 | 29.50 ± 0.46 | 29.59 ± 0.57 | 29.07 ± 0.39 | 27.57 ± 0.35 | 25.02 ± 0.42 |
|  | $\delta$-Targeted[1] | 51.35 ± 0.68 | 50.34 ± 0.60 | 49.27 ± 0.43 | 47.84 ± 1.11 | 46.69 ± 0.84 | 44.81 ± 0.54 |
|  | $\delta$-Targeted[3] | **51.65 ± 0.81** | **51.29 ± 0.33** | **50.74 ± 0.20** | **49.88 ± 0.47** | **48.84 ± 0.61** | **46.95 ± 0.80** |
| $D_t$ | *BEFORE* | 52.17 ± 0.00 | 52.17 ± 0.00 | 52.17 ± 0.00 | 52.17 ± 0.00 | 52.17 ± 0.00 | 52.17 ± 0.00 |
|  | NegGrad | 37.86 ± 0.91 | 41.03 ± 0.91 | 40.44 ± 0.85 | 35.77 ± 1.19 | 29.85 ± 0.83 | 27.48 ± 0.81 |
|  | RandL | 38.51 ± 0.72 | 38.15 ± 0.69 | 38.67 ± 0.63 | 40.35 ± 0.34 | 41.80 ± 0.37 | 42.68 ± 1.00 |
|  | Bad Teacher | 28.60 ± 1.01 | 27.45 ± 2.33 | 28.70 ± 0.88 | 26.14 ± 0.50 | 21.80 ± 0.16 | 21.20 ± 0.37 |
|  | Amnesiac | 29.36 ± 0.65 | 29.38 ± 0.43 | 29.65 ± 0.64 | 28.93 ± 0.43 | 27.40 ± 0.37 | 24.67 ± 0.53 |
|  | $\delta$-Targeted[1] | 50.54 ± 0.80 | 49.73 ± 0.68 | 48.46 ± 0.34 | 46.88 ± 1.11 | 45.77 ± 0.78 | 44.15 ± 0.62 |
|  | $\delta$-Targeted[3] | **50.89 ± 0.90** | **50.68 ± 0.39** | **50.02 ± 0.19** | **48.97 ± 0.60** | **47.86 ± 0.63** | **46.25 ± 0.83** |

**Table 16: Means and standard deviations (percentage) of classification accuracy on Tiny-ImageNet with ViT.**

|  | Method | $\|D_f\|$=16 | $\|D_f\|$=32 | $\|D_f\|$=64 | $\|D_f\|$=128 | $\|D_f\|$=256 | $\|D_f\|$=512 |
|---|---|---|---|---|---|---|---|
| $D_f$ | *BEFORE* | 49.75 ± 1.25 | 49.50 ± 0.50 | 50.05 ± 2.25 | 50.10 ± 0.40 | 51.55 ± 1.45 | 47.00 ± 0.00 |
|  | NegGrad | 0.00 ± 0.00 | 0.00 ± 0.00 | 0.00 ± 0.00 | 0.00 ± 0.00 | 0.00 ± 0.00 | 0.00 ± 0.00 |
|  | RandL | 0.00 ± 0.00 | 0.00 ± 0.00 | 0.00 ± 0.00 | 0.00 ± 0.00 | 0.00 ± 0.00 | 0.00 ± 0.00 |
|  | Bad Teacher | 0.00 ± 0.00 | 0.00 ± 0.00 | 0.00 ± 0.00 | 0.31 ± 0.38 | 0.31 ± 0.29 | 0.35 ± 0.19 |
|  | Amnesiac | 0.00 ± 0.00 | 0.00 ± 0.00 | 0.00 ± 0.00 | 0.00 ± 0.00 | 0.00 ± 0.00 | 0.00 ± 0.00 |
|  | $\delta$-Targeted[1] | 0.00 ± 0.00 | 0.00 ± 0.00 | 0.00 ± 0.00 | 0.00 ± 0.00 | 0.00 ± 0.00 | 0.00 ± 0.00 |
|  | $\delta$-Targeted[3] | 0.00 ± 0.00 | 0.00 ± 0.00 | 0.00 ± 0.00 | 0.00 ± 0.00 | 0.00 ± 0.00 | 0.00 ± 0.00 |
| $D_r$ | *BEFORE* | 54.89 ± 0.00 | 54.89 ± 0.00 | 54.89 ± 0.00 | 54.89 ± 0.00 | 54.89 ± 0.00 | 54.89 ± 0.00 |
|  | NegGrad | 37.60 ± 1.96 | 43.18 ± 5.69 | 44.17 ± 3.90 | 36.36 ± 6.14 | 30.96 ± 4.92 | 29.29 ± 5.16 |
|  | RandL | 36.90 ± 4.11 | 35.76 ± 3.72 | 37.21 ± 4.06 | 40.06 ± 7.68 | 43.20 ± 4.56 | 43.41 ± 5.04 |
|  | Bad Teacher | 28.36 ± 2.01 | 29.74 ± 3.41 | 25.26 ± 1.75 | 25.49 ± 6.03 | 23.50 ± 3.93 | 19.59 ± 3.70 |
|  | Amnesiac | 29.61 ± 3.36 | 27.86 ± 3.56 | 26.82 ± 2.44 | 27.45 ± 5.00 | 28.32 ± 2.86 | 24.63 ± 3.36 |
|  | $\delta$-Targeted[1] | 50.67 ± 2.84 | 49.30 ± 5.14 | 45.94 ± 3.66 | 46.37 ± 2.44 | **49.24 ± 3.04** | 45.51 ± 2.92 |
|  | $\delta$-Targeted[3] | **54.00 ± 7.63** | **49.75 ± 2.40** | **49.18 ± 4.68** | **48.39 ± 4.99** | 48.51 ± 6.30 | **50.47 ± 2.64** |
| $D_t$ | *BEFORE* | 52.34 ± 0.00 | 52.34 ± 0.00 | 52.34 ± 0.00 | 52.34 ± 0.00 | 52.34 ± 0.00 | 52.34 ± 0.00 |
|  | NegGrad | 39.64 ± 4.54 | 37.96 ± 2.64 | 37.58 ± 2.14 | 39.75 ± 2.97 | 30.48 ± 4.31 | 31.76 ± 1.83 |
|  | RandL | 39.02 ± 3.22 | 37.39 ± 3.91 | 35.73 ± 2.03 | 38.69 ± 2.03 | 41.61 ± 4.28 | 44.66 ± 5.27 |
|  | Bad Teacher | 27.57 ± 5.18 | 29.71 ± 2.74 | 31.72 ± 5.41 | 25.34 ± 5.20 | 23.34 ± 3.93 | 21.68 ± 2.21 |
|  | Amnesiac | 27.35 ± 2.90 | 32.21 ± 2.56 | 28.33 ± 4.09 | 32.29 ± 4.37 | 31.35 ± 3.69 | 22.13 ± 4.70 |
|  | $\delta$-Targeted[1] | **51.31 ± 1.81** | 50.13 ± 3.21 | **48.87 ± 7.05** | 47.13 ± 3.20 | 46.59 ± 3.44 | **45.59 ± 3.78** |
|  | $\delta$-Targeted[3] | 49.60 ± 2.88 | **50.42 ± 1.82** | 48.52 ± 5.52 | **52.09 ± 3.99** | **48.85 ± 2.92** | 43.44 ± 2.96 |

## B.5    Changes in Last-Layer Parameters for Different Methods.

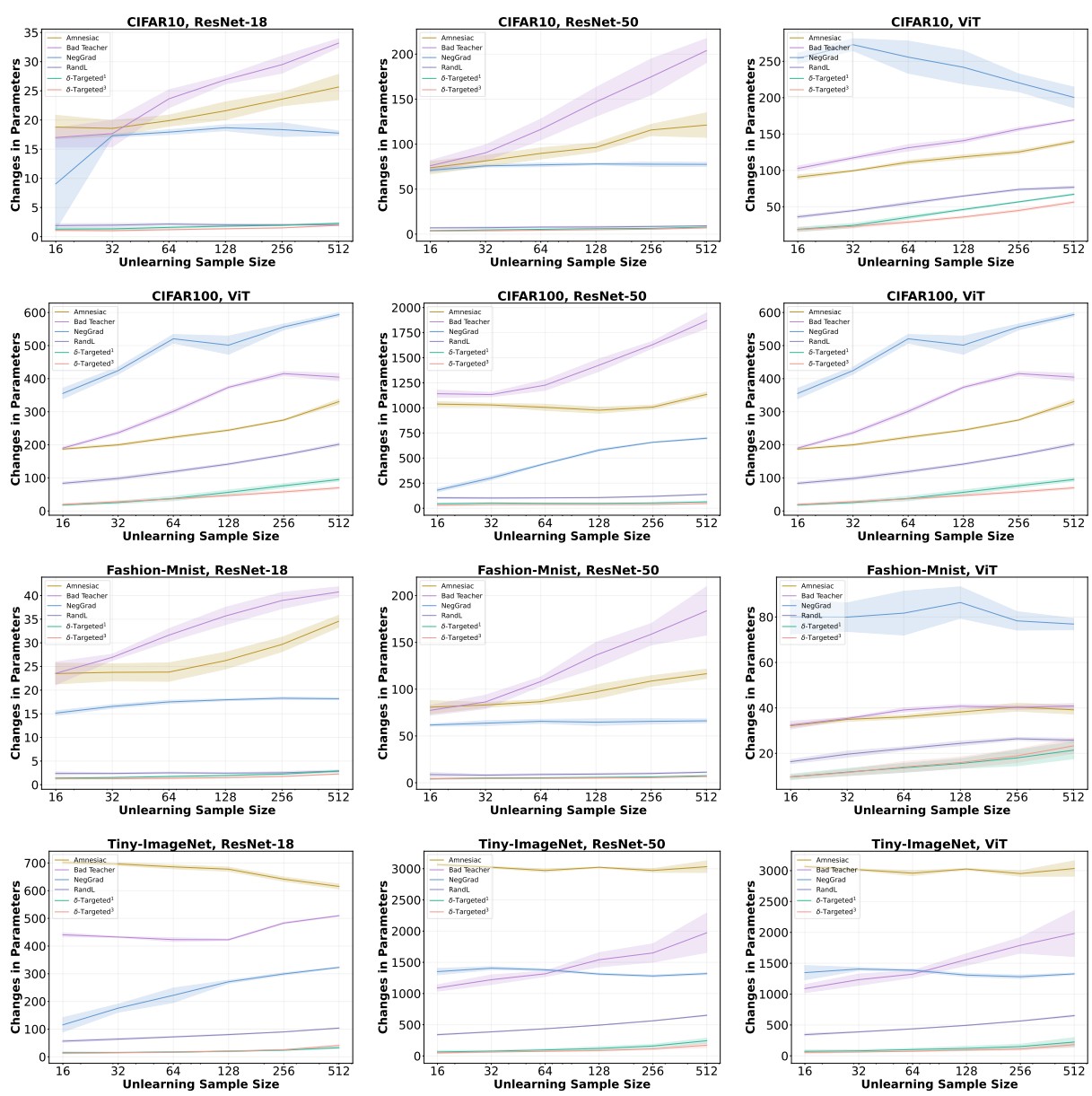

**Figure 4: The magnitude of parameter changes across the model, quantified by the $L_1$-norm. We use $\Delta\theta$ to denote the change in all parameters and $\Delta\phi$ to denote the change in the last layer's parameters. Our methods lead to the smallest change to models' parameters on different datasets.**

## B.6    Changes in All Parameters for Different Methods.

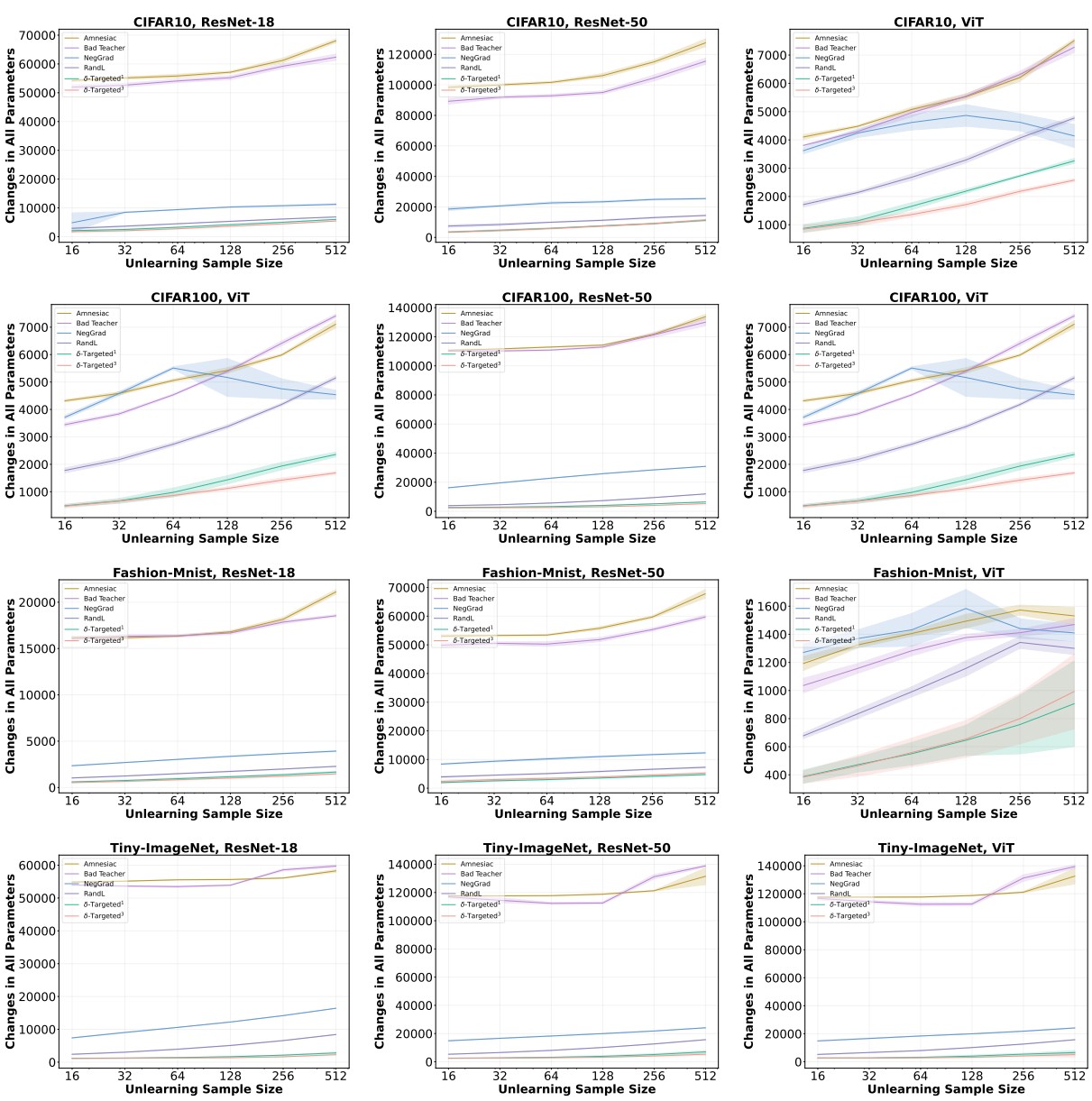

**Figure 5: The parameter changes across different neural network structures. The changes are quantified by the $L_1$-norm. Our method leads to the smallest change to models' parameters on different datasets and cross different neural network structures.**

## B.7 Relations between Changes in Last-Layer Parameters and Changes of All Parameters.

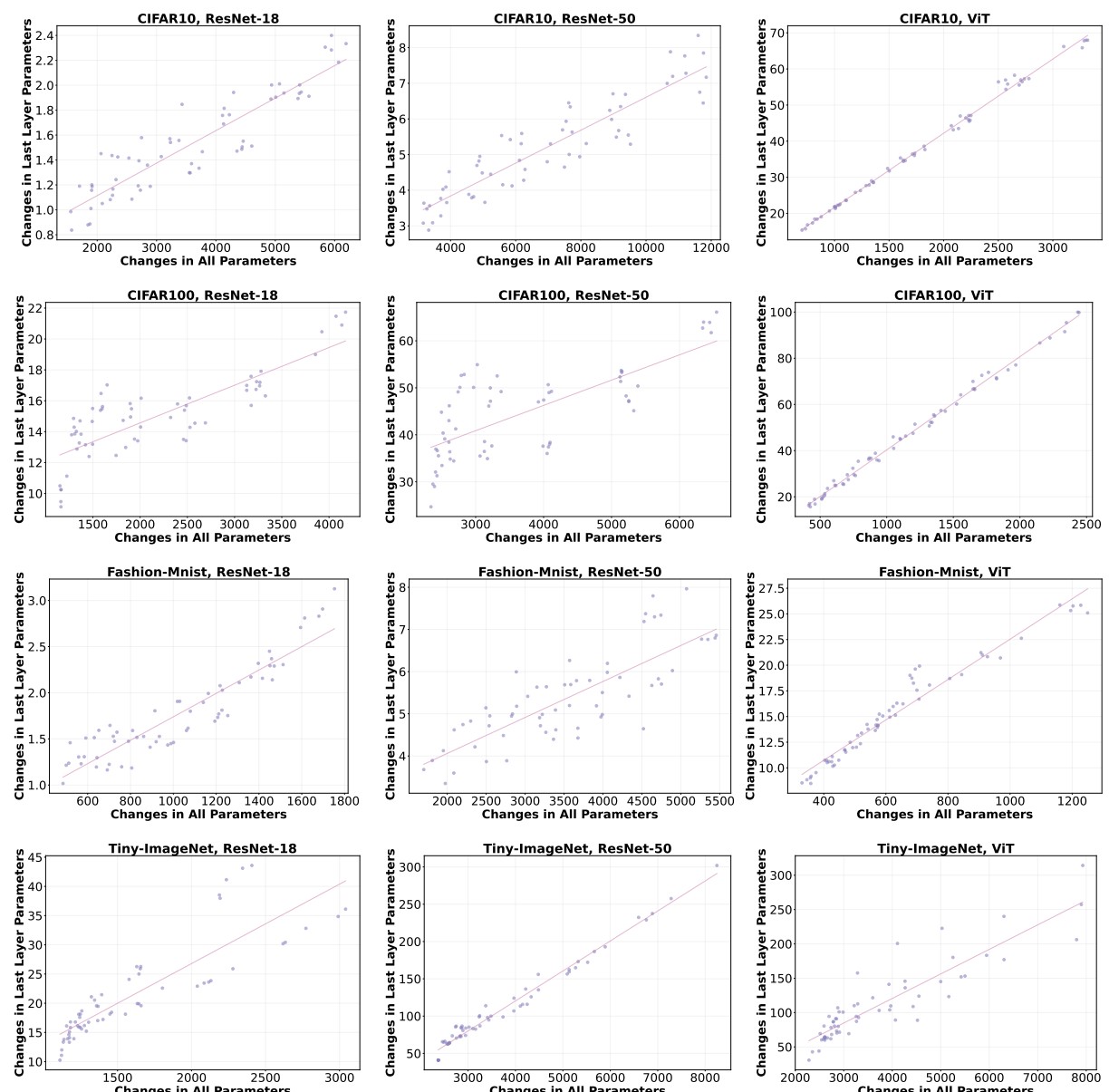

**Figure 6: Dependence between the changes in all parameters and the changes in the last layer parameters by using our method. The magnitude of the parameter changes is quantified by the $L_1$-norm. The result shows a positive correlation.**

## B.8 Relations between a Model's Accuracy on Remaining Examples and Changes of a Model's All Parameters.

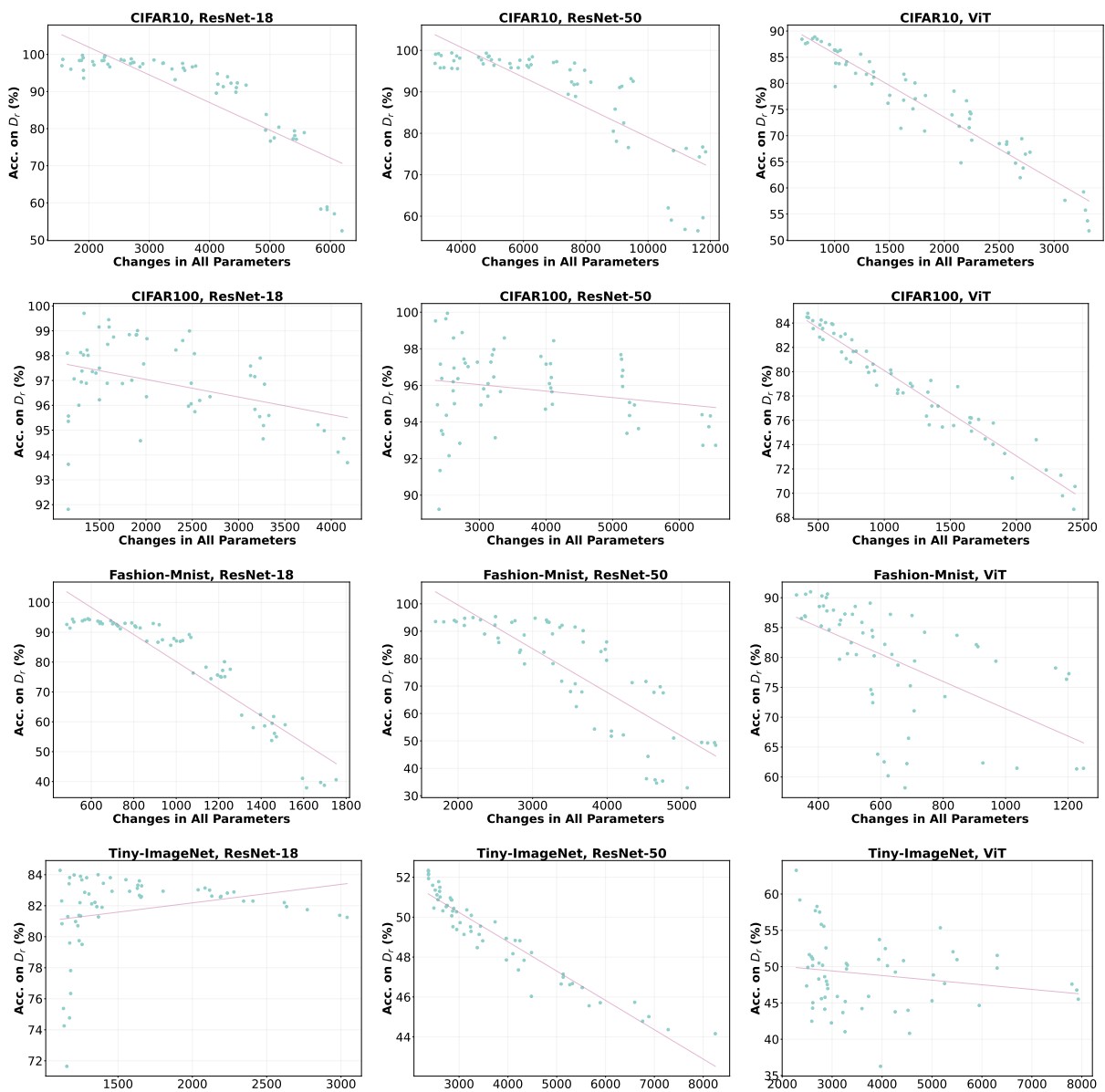

**Figure 7: Dependence between changes in all parameters and model accuracy on remaining examples by using our method. The result shows a negative correlation.**

## B.9 Relations between a Model's Accuracy on Remaining Examples and Changes of a Model's Last-Layer Parameters.

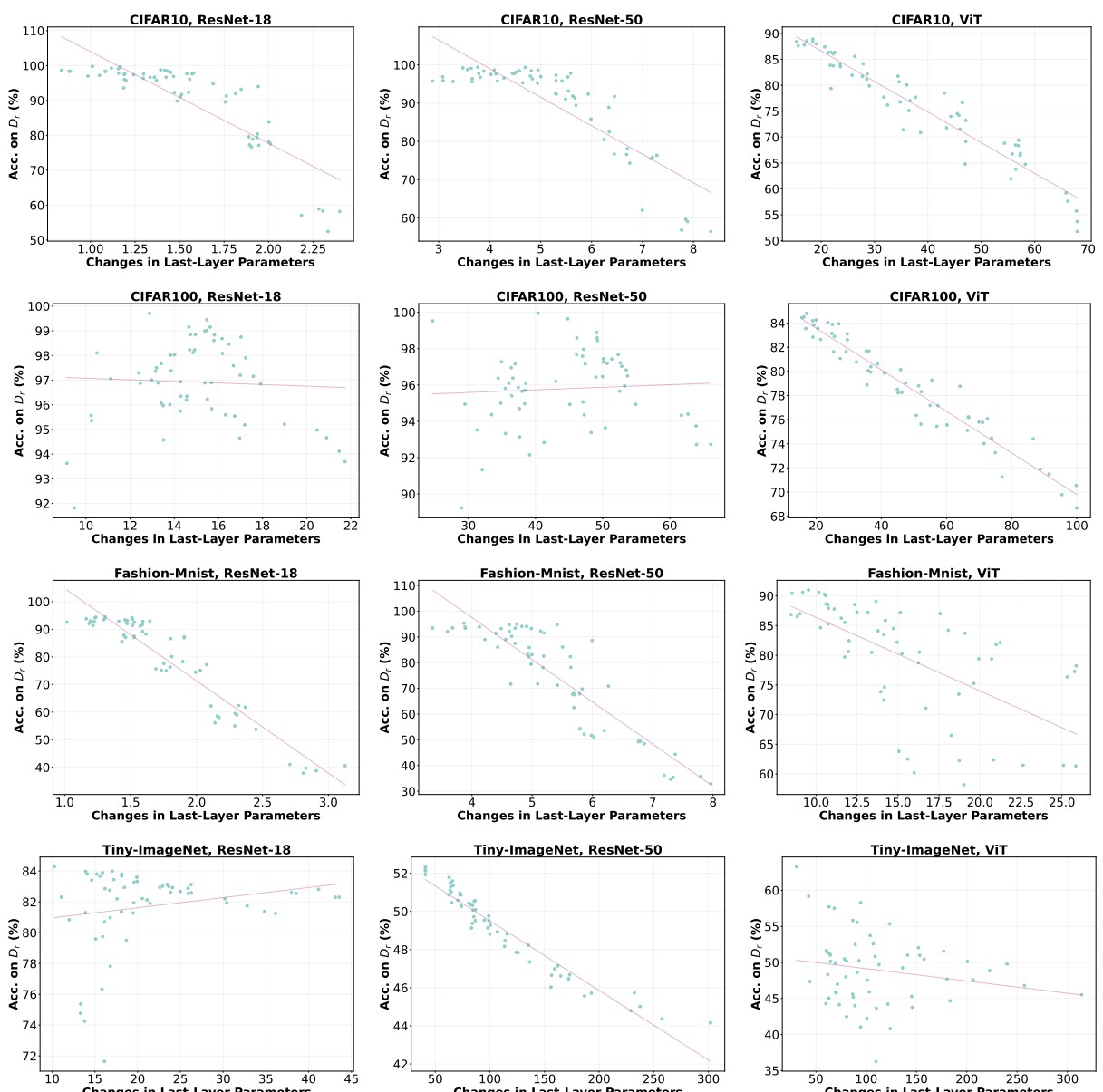

Figure 8: Dependence between changes in last-layer parameters and model accuracy on remaining examples by using our method. The result shows a negative correlation.

