# OpenReview forum: "Towards Safe Machine Unlearning: a Paradigm that Mitigates Performance Degradation"
_ACM.org/TheWebConf/2025/Conference — WWW 2025 Poster_

### Official Review · Reviewer_fcH6 · 2024-12-01

**Novelty:** 4
**Technical Quality:** 4

**Review:**

The paper addresses the important and timely problem of machine unlearning, specifically focusing on mitigating performance degradation while complying with privacy regulations such as the "right to be forgotten." The proposed Targeted Label Noise Injection approach presents an innovative direction by assigning controlled pseudo-labels to the data designated for unlearning, and it claims to achieve minimal performance loss and parameter changes. While the paper explores a critical and under-researched topic, there are several areas where it could benefit from further development to fully establish its contributions and practical impact.

**Questions:**

First, the motivation for the proposed method could be articulated more clearly. While the paper identifies challenges in existing unlearning methods, such as their reliance on remaining training data, the novelty of the proposed approach is not entirely evident. For example, its reliance on pseudo-labels and targeted parameter updates resembles techniques already documented in the literature, such as adversarial perturbation or label flipping. Highlighting how this approach uniquely addresses the limitations of current methods, beyond reducing parameter changes, would strengthen the case for its innovation.

Second, there are methodological aspects that require more clarity and justification. The approach assumes the reliability of confidence estimates from the model for determining pseudo-labels, which may not always hold, particularly in noisy or complex data settings. Additionally, the paper introduces a critical hyperparameter (
δ
δ) for controlling the strength of unlearning, but it does not provide sufficient analysis of how this parameter impacts performance across diverse scenarios. A more detailed exploration of the method’s robustness to hyperparameter settings and dataset variations would make the approach more convincing.

Third, the experimental evaluation, while thorough in testing standard datasets like CIFAR-10 and CIFAR-100, could benefit from broader scope. The datasets chosen may not fully reflect the diversity and scale of real-world unlearning scenarios, such as unstructured data, large-scale models, or applications with dynamic data distributions. Furthermore, the comparison with baselines appears uneven, as some methods are granted access to remaining training data while others are not, which complicates a fair assessment. Incorporating metrics like computational efficiency and time complexity, which are critical for practical deployment, would provide a more comprehensive evaluation.

Finally, while the theoretical analysis is rigorous, it might be challenging for readers to connect it to the empirical claims of performance improvement. Simplifying the presentation of these results or illustrating them with intuitive examples could enhance accessibility. Additionally, the figures and tables, while informative, could be streamlined to make key insights more prominent. There are also minor typographical and grammatical inconsistencies, which, while not major, detract from the overall polish of the paper.

**Reviewer Confidence:**

4: The reviewer is certain that the evaluation is correct and very familiar with the relevant literature

**Scope:**

3: The work is somewhat relevant to the Web and to the track, and is of narrow interest to a sub-community

---

### Official Review · Reviewer_bTwy · 2024-12-02

**Novelty:** 5
**Technical Quality:** 5

**Review:**

This paper studied on the problem of machine unlearning, which unlearns specific training data from a pre-trained model without accessing the remaining training data and protect model performance without dramatically changing the model’s parameters. The authors propose a practical method called Targeted Label Noise Injection, which is proved through several experiments on four datasets.

Pros:
1. The authors theoretically prove the effectiveness of the proposed method.
2. They empirically show that it achieves state-of-the-art unlearning performance across various datasets.

Cons:
1. The writing is not clear and difficult to understand, especially symbols.

**Questions:**

1. Please present the meaning of symbols in the problem setup of Section 3.
2. The cases at lines 7 and 11 in Algorithm 1, can be merged.
3. What is the meaning of $\nabla$ L at line 20 in Algorithm 1?
4. In Fig.1, it is difficult to discriminate the results of models. The colors are similar.
5. Why are so many zero values in the D_f part of Table 1?
6. Why does the performance of proposed method fluctuate on four datasets, e.g. for D_t, from 85 to 51?
7. There are many typos or missing, e.g., line 397 degradtion, missing the period at line 397, 690

**Reviewer Confidence:**

2: The reviewer is willing to defend the evaluation, but it is likely that the reviewer did not understand parts of the paper

**Scope:**

3: The work is somewhat relevant to the Web and to the track, and is of narrow interest to a sub-community

---

### Official Review · Reviewer_hrwF · 2024-12-02

**Novelty:** 5
**Technical Quality:** 4

**Review:**

This paper addresses a classic problem in machine learning: unlearning specific data points without access to the remaining dataset while maintaining the model's performance. The authors propose an innovative Targeted Label Noise Injection approach, supported by theoretical and empirical evidence, achieving state-of-the-art results. However, while the methodology and results are promising, the paper's explanations, especially in mathematical formulations and parameter tuning, could benefit from more clarity and contextual grounding.

+ The proposed Targeted Label Noise Injection method is a great contribution, mitigating the model performance drop during unlearning without requiring remaining data access.
+ The paper includes theoretical guarantees, such as the concentration of parameter updates, which align well with the empirical results.
+ Extensive experiments on diverse datasets and general architectures (ResNet-18, ResNet-50, ViT) validate the method's effectiveness.
+ The study directly addresses GDPR-aligned privacy concerns, showcasing applicability in real-world scenarios.

- Some notations (\delta, \lambda) are used with multiple interpretations across sections, leading to potential confusion about their specific roles and parameter tuning strategies.
- While the proposed method outperforms baseline methods, the baselines are somewhat outdated (e.g., "Bad Teacher") and lack comparisons with more recent works in private machine learning or federated learning contexts.
- The claim that \lambda-exponent reduces overfitting is empirically supported but lacks deeper theoretical backing or robust discussion on potential edge cases.
- Limited discussion on the sensitivity of key parameters (\delta, \lambda, \eta) and their interaction could make reproducibility challenging.

**Questions:**

- In Section 3.1, you use the second most confident label as the easy-to-learn pseudo-label. Could this choice introduce systematic bias for certain classes with naturally higher probabilities for second-place predictions?
- How does the choice of \lambda affect the trade-off between accuracy retention and unlearning effectiveness? Theoretical explanations remain qualitative without quantitative guidelines.
- The theorem discusses concentrated parameter updates but does not explicitly relate them to changes in earlier layers of deep networks. How does this localized update propagate in backpropagation-heavy architectures like ViT?
- Could adversaries exploit the predictable nature of pseudo-label assignments to reconstruct sensitive information? How robust is the method under adversarial settings?
- While results on CIFAR and Tiny-ImageNet are convincing, how would the method perform on larger datasets (such as ImageNet) or NLP tasks?

**Reviewer Confidence:**

3: The reviewer is confident but not certain that the evaluation is correct

**Scope:**

3: The work is somewhat relevant to the Web and to the track, and is of narrow interest to a sub-community

---

### Official Review · Reviewer_P2de · 2024-12-02

**Novelty:** 5
**Technical Quality:** 3

**Review:**

This work introduces a new method for unlearning an arbitrary subset of training data without using the original training data to retain model performance. The general idea is adding targeted label noise on the data that needs to be forgotten (forget set). The authors propose using the second most likely (most likely if prediction is wrong) prediction of the model as the new pseudo label as they find that this will change the model parameters the least while still forgetting (predict incorrectly on) these examples. The authors also add 2 hyperparameters, $\alpha$ which controls how each example is dynamically weighted to stop learning from it when the loss is small and $\delta$ which controls how often the label should be perturbed.

In the following section, the authors provide and prove a theorem showing that this update (compared to gradient ascent) provides a more concentrated update.  Then, the authors conduct some experiments to show that this method produces smaller changes to the model parameters compared to others tested.

Finally, the authors provide an experimental study of the algorithm and compare it to 4 others: “NegGrad”, "RandL", "Bad Teacher" and "Amnesiac" on 4 datasets: CIFAR-10, CIFAR-100, Fashion-MNIST, and TinyImageNet using model architectures ResNet18, ResNet50 and ViT. They find that their method outperforms all these methods significantly, including "Bad Teacher" and "Amnesiac" which seem to require some examples from the original training dataset. They also find that smaller parameter updates correspond well with higher accuracy on the original training and test datasets after unlearning and that norm of the update on the last layer is correlated well with the overall norm of the update.

**Questions:**

1) For the proof of Theorem 4.1, it was not clear to me why the second most likely predicted label should produce the most concentrated update. The proof seems to indicate that the least likely label should produce the most concentrated update unless I am missing something. Also, given that you are looking at concentration, is there any notion of how much more concentrated these updates will be?

2) Comparing the theoretical results with the experiments, it seems the theoretical results are looking at concentration at a fixed norm whereas the experiments are correlating smaller parameter updates to better utility on the training dataset. Do you have any experiments which show that your method produces more concentrated updates and that this is better for utility?

3) It's not clear why $\delta$ is needed at all. The theorem and all the experiments set it to 0. It should either be removed to simplify the algorithm or there should be some analysis provided on how it changes the results of the algorithm. Also, can the authors clarify this statement from the paper: "Note that to make the pre-trained model randomly guess a to-be forgotten example, $\delta$ should be set to 1/C, where C is the number of classes"?

4) This might be outside of the threat model but if you had access to the logits, could you reconstruct the original labels from the model? If I had to guess, the second most likely prediction after the unlearning process if likely the true label.

**Reviewer Confidence:**

3: The reviewer is confident but not certain that the evaluation is correct

**Scope:**

3: The work is somewhat relevant to the Web and to the track, and is of narrow interest to a sub-community